# Novel Dead-Time Compensation Strategy for Wide Current Range in a Three-Phase Inverter

**Jeong-Woo Lim, Hanyoung Bu and Younghoon Cho *** 

Department of Electrical Engineering, Konkuk University, Seoul 05029, Korea; wjddn1@konkuk.ac.kr (J.-W.L.); bhy1014@konkuk.ac.kr (H.B.)
* Correspondence: yhcho98@konkuk.ac.kr; Tel.: +82-10-6207-0431

**Abstract:** This paper proposes a novel three-phase voltage source inverter dead-time compensation strategy for accurate compensation in wide current regions of the inverter. In particular, an analysis of the output voltage distortion of the inverter, which appears as parasitic components of the switches, was conducted for proper voltage compensation in the low current region, and an on-line compensation voltage controller was proposed. Additionally, a new trapezoidal compensation voltage implementation method using the current phase was proposed to simplify realizing the trapezoidal shape of the three-phase compensation voltages. Finally, when the proposed dead-time compensation strategy was applied, the maximum phase voltage magnitude in the linear modulation voltage regions was defined to achieve smooth operation even at high modulation index. Simulations and experiments were conducted to verify the performance of the proposed dead-time compensation scheme.

**Keywords:** dead-time compensation strategy (DTCS); dead-time compensation; trapezoidal compensation voltage; dead-time effects; three-phase voltage source inverter (VSI) compensation

## 1. Introduction

Dead-time is an efficient strategy which adds blank time within complementary switching signals to prevent arm-short. The series two switches circuit, sharing a DC-link such a half-bridge, is operating complementary to avoid arm-short condition. However, in the actual switch, a delay occurs within on/off operating due to the parasitic components, and the series switches appear to be shorted with a DC-link. The short circuit allows excessive current through the series switches, causing serious system failure. Therefore, the reliability of the system can be guaranteed by injecting enough dead-time ($T_d$) until the switch reaches a steady state [1,2].

Especially, as shown in Figure 1a, a circuit structure such as a typical three-phase VSI in which three legs share a DC-link must ensure reliability of the system by applying dead-time. Figure 1b shows the operation of a single leg over time during one switching period. $Q_{1s}^*$, $\overline{Q}_{1s}^*$ are ideal complementary switches on/off signals, $Q_{1s}$, $\overline{Q}_{1s}$ are real switch on/off signals adapted to the dead-time $T_d$ and the subscripts $_{a,b,c}$ indicate each phase. For example, $i_a, i_b, i_c$ are a-, b- and c-phase current and $R, L, e$ are phase resistor, inductor and voltage source, respectively. Since the dead-time cannot control actively, it causes not only serious voltage distortions in inverter output voltage as shown Figure 1b, but also adverse effects on the all algorithms using a voltage reference [3,4]. Figure 2a shows a pole voltage reference of a-phase in the three-phase VSI and (Figure 2b) shows current waveforms applied the dead-time to compare the ideal current waveform. Here, in the current waveform to which the dead time is applied, it can be seen that serious current distortion occurs near the zero point and near the peak area. Various types of dead-time compensation strategies have been published to analyze and compensate for the dead-time defects. In [1,2,5], the dead-time and the switch on/off delay were analyzed and the dead-time compensation method via ideal parameters was suggested. However, since

the switch parameters are fluctuated with external factors, it is difficult to compensate accurately in all inverter operating areas by using fixed variables. In some papers [6–8], the distortions of inverter output voltage by switch's output capacitors were studied and compensation strategies with the look-up table containing the switch-off times according to the magnitude of the current were suggested. Although, a disadvantage is that it is difficult to compensate the precise dead-time in various environments since the table is limited to the experimental environment. The papers [9,10] proposed an on-line dead-time compensation method which modifies the dead-time compensation voltage (DTCV) by feeding back current distortions. However, the strategies extracting the current distortions are complicated, and have drawbacks near the current zero-crossing points. In [11], the dead-time compensation algorithm using a filter has been suggested. However, due to the lowpass-filter characteristics, the bandwidth of the current controller can be limited. In [12], a scheme which compensates the sixth-order harmonic in *d-q* axis currents on the synchronous reference frame using a bandpass filter is suggested. However, the performance of the dead-time compensation algorithm is limited by the characteristic of the current controller, making it difficult to compensate for all the dead-time effects of wide harmonics. References [13,14] offer compensation strategies using an observer which is feeding back *d-q* axis currents on synchronous reference frame. However, since it is utilizing an ideal-model, the observer regards not only the dead-time distortions but also various parasitic components as dead-time errors. Thus, it is impossible to accurately estimate the real output voltage of the inverter. In [15,16], the on-line dead-time compensation algorithms having a trapezoidal shape compensation voltage and a modulator for slope have been proposed. However, it is difficult to completely compensate for the non-linearities of the switch, especially in the low current region. The repetitive controller for dead-time compensation is proposed. Since it is based on the permanent magnet synchronous motor (PMSM) position not constant time like the conventional repetitive controller, it is robust at the variation of the motor speed. However, the d-axis current on the synchronous reference frame affects to the position information, and as a result, precise dead-time compensation is impossible [17]. A method to compensate the output voltage error of the inverter using the information of the terminal voltage of the inverter is suggested, but it requires additional hardware to sense the terminal voltage, so it is difficult to apply it to the existing three-phase VSI which may lead to an increase in cost [18,19].

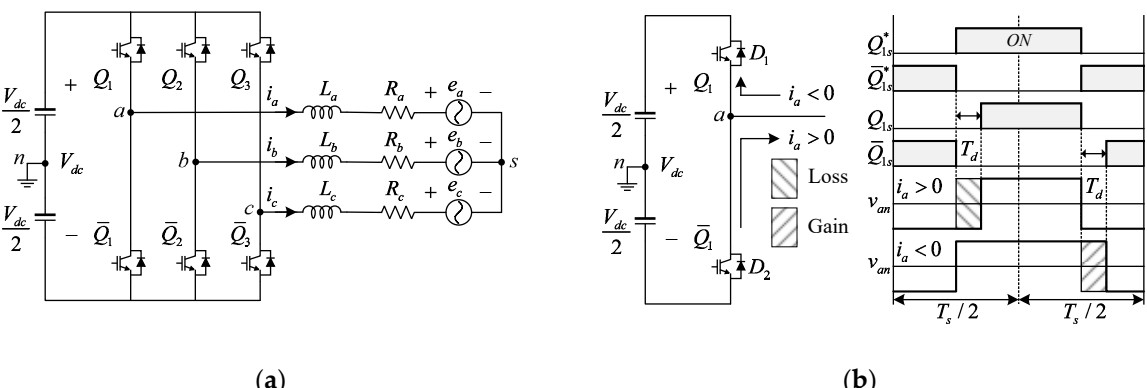

(a)                    (b)

**Figure 1.** Three-phase VSI and dead-time switching patterns of a-phase leg: (**a**) Typical three-phase VSI configuration; (**b**) switching patterns and current flow direction of the one phase leg during the dead-time.

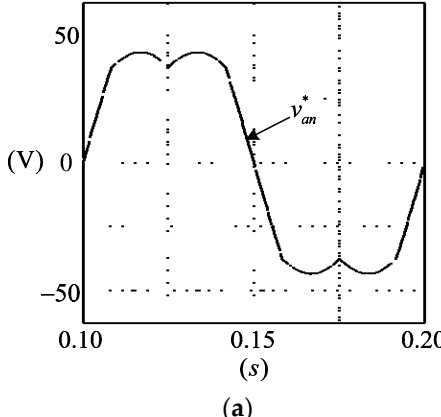
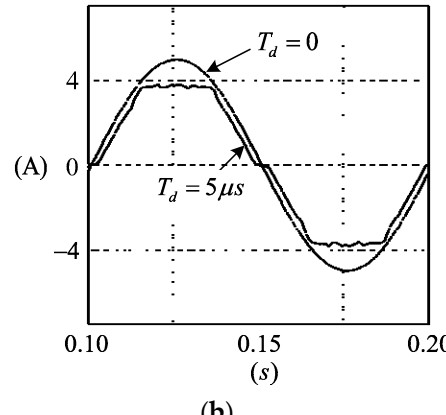

**Figure 2.** Effects of the dead-time in Figure 1a ($f_{sw}$ = 20 kHz, $V_{dc}$ = 100 V, $L_s$ = 0.01 mH, $R_s$ = 10 $\Omega$); (**a**) an a-phase pole voltage reference as space vector PWM (SVPWM); (**b**) comparing dead-time effects with equal voltage reference with (**a**).

In this paper, a novel DTCS for accurate dead-time compensation in all output regions of the inverter is proposed. In particular, with the dead-time compensation algorithm using passive calculation it is difficult to accurately compensate all areas of the inverter output because increased switch turn-off delay effects occur due to the parasitic components of the switch in the low current region. Therefore, in this paper, a new controller that can actively compensate for voltage distortion due to the switch parasitic components and a new method that can more easily implement the three-phase trapezoidal compensation voltage (TCV) is presented. In Section 2, the inverter output voltage error by the dead-time and the switch's non-linearities are analyzed. In Section 3, a novel three-phase TCV implementation strategy and the on-line TCV controller revising its amplitude and slope is presented. In Section 4, the output voltage distortions of the three-phase VSI and the maximum linear-modulation phase voltage (MMPV) are analyzed on the space vector area. Finally, in Section 5, the simulation and the experiment are implemented to verify the proposed DTCS. The performance of the DTCS is evaluated with phase current total harmonic distortion (THD).

## 2. Analysis of the Dead-Time Effects

In this paper, the dead-time $T_d$ of Equation (2) includes the ON/OFF propagation delay in order to be simply expressed as $V_d$. Additionally, the conduction voltage drops across the diode and switch are excluded from the effect of dead-time because they are negligible compared to the DC-link voltage level.

### 2.1. The Three-Phase VSI Output Voltage Errors by the Dead-Time

In Figure 1b, an a-phase single leg output voltage $v_{an}$ is varied according to the phase current $i_a$ direction during the dead-time. When the current direction is positive, the current flows through the body diode $D_2$ in the lower switch $\overline{Q}_1$, so that the $v_{an}$ comes to be $-V_{dc}/2$. On the other hand, when the current direction is negative, the current flows through the body diode $D_1$ in the upper switch $Q_1$, thus the output $v_{an}$ becomes $V_{dc}/2$. Therefore, the a-phase pole voltage errors due to the dead-time can be expressed as

$$\Delta v_{an}^{err} = \begin{cases} -V_d & (i_a > 0) \\ V_d & (i_a < 0) \end{cases}, \tag{1}$$

$$V_d = \frac{T_{on} + T_d - T_{off}}{T_s} V_{dc}. \tag{2}$$

In Equation (2), the $V_d$ is average pole voltage error (APVE) that occurs during a single switching period, and it contains switch turn on/off delays $T_{on}, T_{off}$ as well as dead-time $T_d$ [2]. The a-phase APVE can be expressed according to the direction of current as shown Equation (1). In addition, the

other phases, b and c, can be expressed in the same approach via each current polarity [1]. The APVEs of three-phase can be represented by the voltage errors on the synchronous reference frame *d-q* axis as follows.

$$
\begin{bmatrix} \Delta v_{as}^{err} \\ \Delta v_{bs}^{err} \\ \Delta v_{cs}^{err} \end{bmatrix} = \frac{1}{3} \begin{bmatrix} 2 & -1 & -1 \\ -1 & 2 & -1 \\ -1 & -1 & 2 \end{bmatrix} \begin{bmatrix} \Delta v_{an}^{err} \\ \Delta v_{bn}^{err} \\ \Delta v_{cn}^{err} \end{bmatrix},
\tag{3}
$$

$$
\begin{bmatrix} \Delta v_d^{err} \\ \Delta v_q^{err} \end{bmatrix} = \begin{bmatrix} \cos\theta_e & \sin\theta_e \\ -\sin\theta_e & \cos\theta_e \end{bmatrix} \begin{bmatrix} 1 & 0 & 0 \\ 0 & \frac{1}{\sqrt{3}} & -\frac{1}{\sqrt{3}} \end{bmatrix} \begin{bmatrix} \Delta v_{as}^{err} \\ \Delta v_{bs}^{err} \\ \Delta v_{cs}^{err} \end{bmatrix}
$$

$$
= -\frac{4V_d}{\pi} \begin{bmatrix} -\sin\delta - \sum_{n=1}^{\infty} \left\{ \frac{\sin(6n\omega_e t - \delta)}{(6n-1)} + \frac{\sin(6n\omega_e t + \delta)}{(6n+1)} \right\} \\ \cos\delta - \sum_{n=1}^{\infty} \left\{ \frac{\cos(6n\omega_e t - \delta)}{(6n-1)} - \frac{\cos(6n\omega_e t + \delta)}{(6n+1)} \right\} \end{bmatrix}
\tag{4}
$$

Equation (3) is expressed as the average phase voltage error by APVE [2], and Equation (4) represents the average phase voltage error due to the dead-time on the synchronous reference frame *d-q* axis by the Fourier series expansion [5,11], where $\theta_e, \omega_e$ are electrical angle, electrical angular velocity, $i_d, i_q$ are *d-*, *q*-axis current respectively and $\delta$ is the phase angle between *q*-axis and three-phase current vector $I_s$ as shown Figure 3.

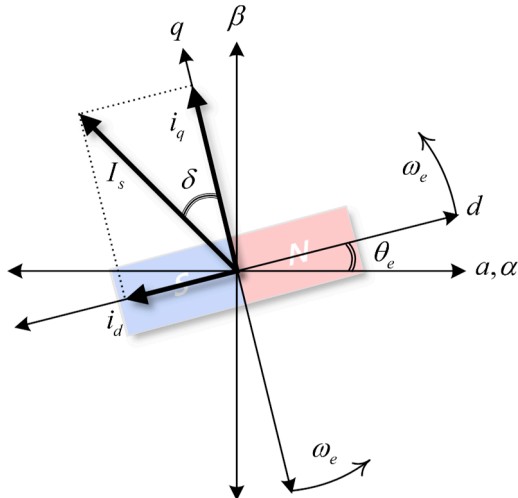

**Figure 3.** Stationary reference frame $\alpha - \beta$ axis and synchronous reference frame $d - q$ axis with $\delta$.

The *d-q* axis voltage error contains both the fundamental and the 6*n*th harmonics distortions, as in Equation (4). Theses voltage errors cause discordance between the real output voltage of three-phase VSI and the voltage commands. Furthermore, the distortion components cause harmonic currents, which degrade the performance of the VSI. Therefore, in order to compensate the voltage distortions due to the dead-time, the opposite voltages of the error voltages can be generated using the Equation (1). The average compensation pole voltage (ACPV) can be expressed as follows.

$$
\Delta v_{an} = V_d \mathrm{sign}(i_a),
\tag{5}
$$

$$
\mathrm{sign}(i_a) = \begin{cases} 1 & (i_a > 0) \\ -1 & (i_a < 0) \end{cases}.
\tag{6}
$$

The ACPVs of the b- and c-phase can be described in a similar way as Equations (5) and (6) which is a-phase AVPC [1,2]. Using the three-phase ACPVs and *d-q* axis transformation matrix (4), the *d-q* axis dead-time compensation voltage waveforms can be illustrated as Figure 4. Figure 4a shows the a-phase current and Figure 4b,c reveals the average compensation pole voltage of the a-phase and the

average compensation phase voltage of the a-phase, respectively. Figure 4d presents the dead-time compensation voltage converted to the $\alpha - \beta$ axis, and Figure 4e shows the dead time compensation voltage on the *d-q* axis when $\delta = 0$.

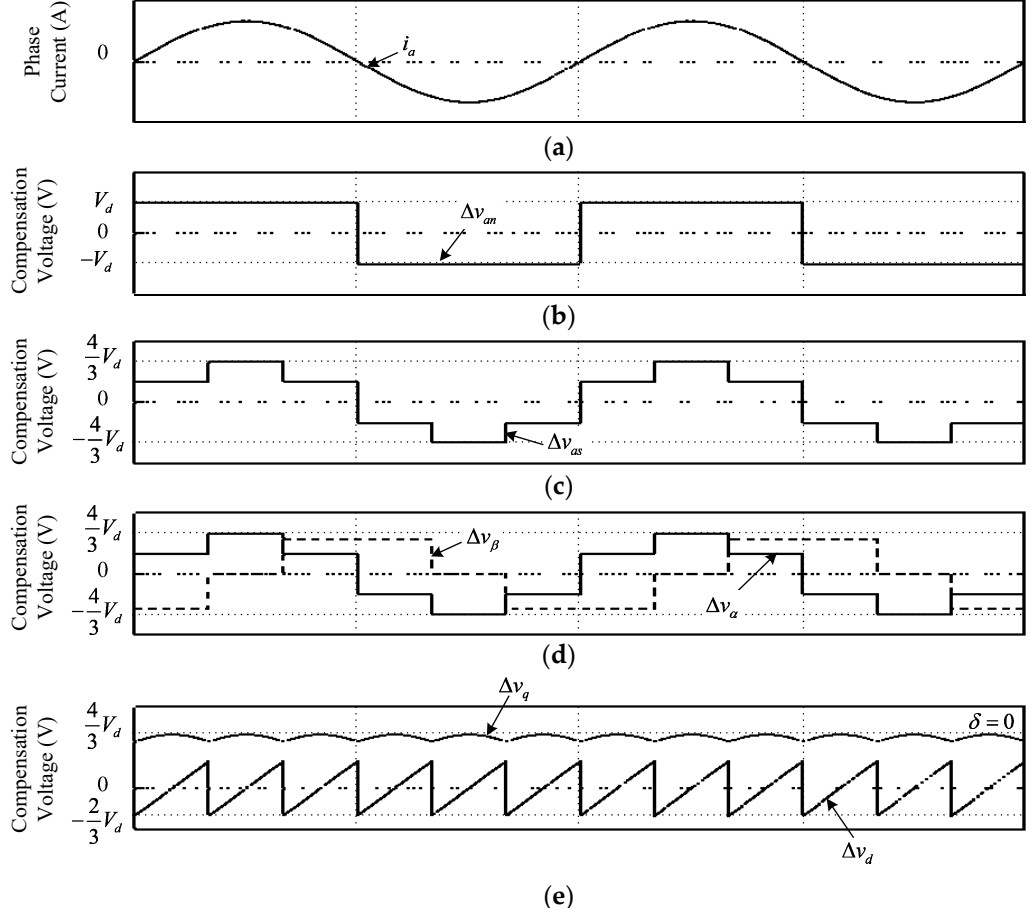

**Figure 4.** Dead-time compensation voltage waveforms; (**a**) a-phase current; (**b**) average compensation pole voltage (ACPV) of a-phase; (**c**) the average compensation phase voltage of a-phase; (**d**) dead-time compensation voltages on stationary reference frame $\alpha - \beta$ axis; (**e**) dead-time compensation voltages on the synchronous reference frame $d - q$ axis.

## 2.2. The Effects of Switch Parasitic Components

The real switch contains diverse parasitic components and the output capacitor of the switch is a critical factor in compensating the distorted output voltage of three-phase VSI because it seriously affects the switch off delay time, $T_{off}$, depending on the magnitude of the phase current [6]. Figure 5 exposes the output capacitors $C_1, C_2$ connected in parallel with the switches and the charging and discharging process when the phase current $i_p$ flows in the positive direction.

In Figure 5a, the dc-link voltage $\left(v_{C_1} = V_{dc}\right)$ is charging to the capacitor $C_1$ while the upper switch is turning on. At that moment, the discharging current $-i_{C_1}$ flows to the node 'p' due to the potential difference and is charging the capacitor $C_2$ of the lower switch so that the voltage $v_{C_2}$ of the lower switch parasitic capacitor $C_2$ is rapidly charged to $V_{dc}$. Consequently, when the upper switch is turning on, the output pole voltage $v_{pn}$ of the half-bridge is hardly affected. On the contrary, when the both upper and lower switches are turned off (during the dead-time), as shown in Figure 5b, the capacitor $C_2$ of the lower switch is discharged and the voltage of $v_{C_2}$ arrives at zero. At this time, the current $i_p$ can

be expressed by the sum of the charging current $i_{C_1}$ of the upper output capacitor and the discharging current $i_{C_2}$ of the lower output capacitor.

$$i_p = -i_{C_1} + i_{C_1}. \tag{7}$$

Assuming that the capacitances $C_1, C_2$ and the charging/discharging potentials are equal, the turn off delay time $T_{off}$ required for discharging $v_{C_2}$ can be formulated as follows.

$$C_1 = C_2 = C_{12}, \tag{8}$$

$$\left| i_{C_2} \right| = \left| i_{C_1} \right| = \frac{\left| i_p \right|}{2}, \tag{9}$$

$$v_{C_2}\left(T_{off}\right) = \frac{1}{C_{12}} \int_0^{T_{off}} -i_{C_2}(t)dt + v_{C_2}(0). \tag{10}$$

From the Equation (10), when the initial value is $V_{dc}$, the following Equations (11), (12) can be obtained.

$$-V_{dc} = \frac{1}{C_{12}} \int_0^{T_{off}} -\frac{\left| i_p \right|}{2} dt, \tag{11}$$

$$\therefore T_{off} = \frac{2C_{12}V_{dc}}{\left| i_p \right|} \quad \left(T_d \geq T_{off}\right). \tag{12}$$

If the current $i_p$ flows in the opposite direction, the switch turn off delay occurs in the upper switch in a similar way when the current is flowing in the positive direction, as shown Figure 5. Thus, the upper switch turns off delay is the same as in Equation (12) [20]. Figures 6 and 7 show the simulation waveforms and the results verify the previously defined equations in Section 2.2. The graph of Figure 6 compares the simulation results of Figure 7 with Equation (12), where $T_{off}^{sim}$ is the turn off delay times measured using the simulation results in Figure 7, and $T_{off}^{eq}$ is the turn off delay times calculated using the Equation (12) respectively. Here, the simulation circuit configuration of Figure 7 is arranged such as Figure 5a, and the lower switch is maintaining the turning off condition while the upper switch is turning on and off. Figure 7a shows the gate-source voltage of the upper switch, and the lower switch waveform is omitted because it only applies the off signal. Figure 7c displays the pole voltage, and voltage distortion due to the output capacitor can be confirmed.

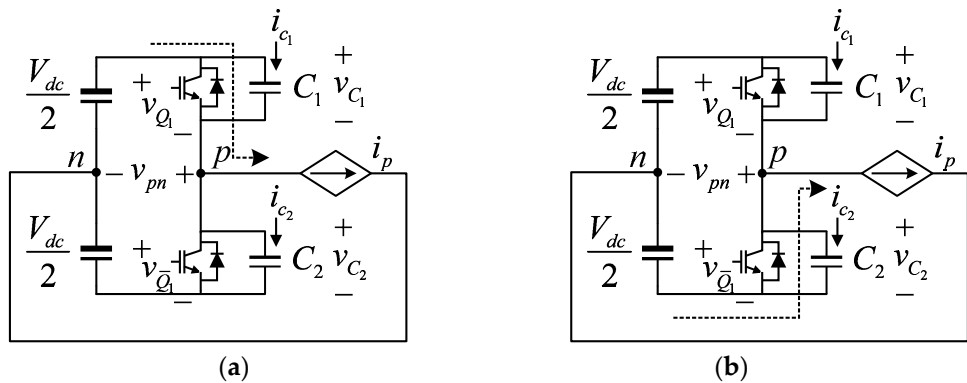

**Figure 5.** Charging and discharging process of the output capacitors $\left(i_p > 0\right)$; (**a**) the upper switch is turning on; (**b**) the upper switch is turning off.

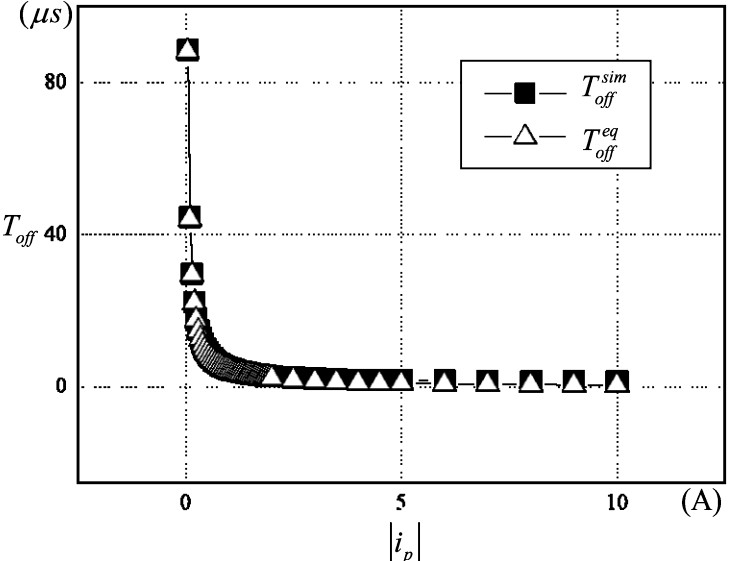

**Figure 6.** Comparing $T_{off}^{eq}$ with $T_{off}^{sim}$.

Figure 6 demonstrates that the turn-off delay time $T_{off}$ increases very nonlinearly with current magnitude. The characteristics of these output capacitors indicate that compensation strategies that can be actively controlled by considering switch parasitic components are essential for accurate dead-time compensation in wide current ranges.

When the current $i_p$ is positive direction, the output pole voltage of the half-bride represented as

$$v_{pn} = -\frac{V_{dc}}{2} + v_{C_2}. \tag{13}$$

Here, the $v_{C_2}$ affects the output of the inverter since it is discharged with a slope depending on the level of $|i_p|$ as shown in Figure 7. Therefore, the $v_{C_2}$ should be properly compensated because it cannot be actively controlled.

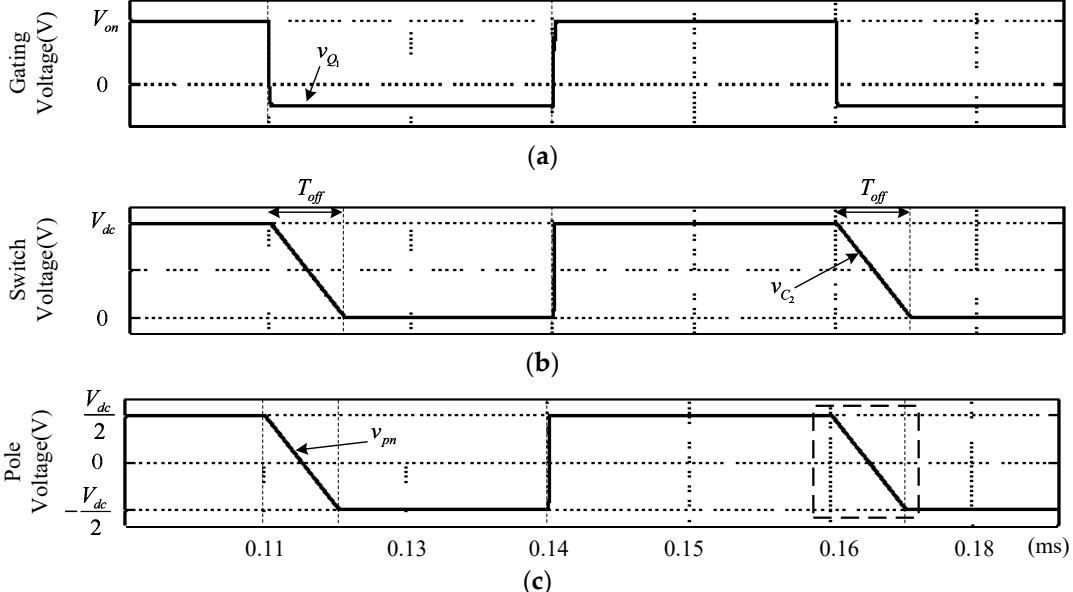

**Figure 7.** Simulation results of Figure 5; (**a**) gate voltage $v_{Q_1}$; (**b**) lower output capacitor voltage $v_{C_2}$; (**c**) inverter pole voltage $v_{pn}$.

Figure 8 shows the actual waveform of $v_{C_2}$, which varies with current magnitude, compared with Figure 7, which is the simulation result.

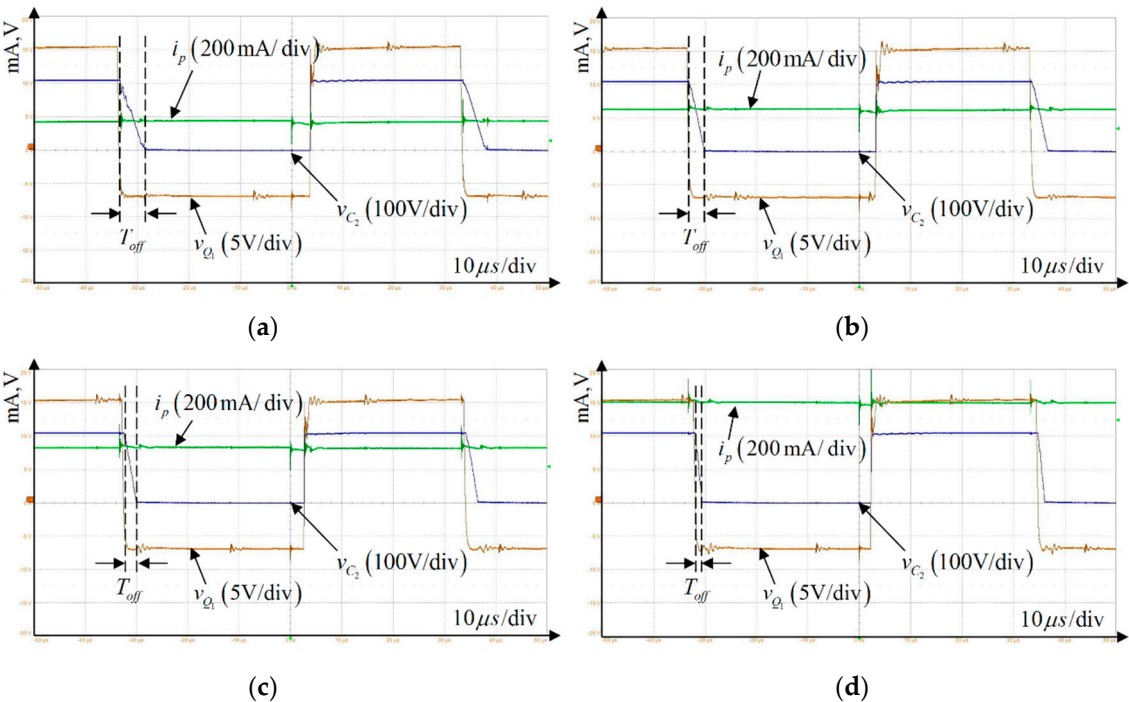

(a)

(b)

(c)

(d)

**Figure 8.** Variation of $T_{off}$ according to the current $i_p$ magnitude ($R_{on}$ = 23.5 Ω $R_{off}$ = 23.5 Ω); (**a**) $i_p$ = 170 mA; (**b**) $i_p$ = 250 mA; (**c**) $i_p$ = 330 mA; (**d**) $i_p$ = 600 mA.

Equation (2) is applicable when the current level is enough to saturate $T_{off}$ and the voltage of the output capacitor is rapidly falling or rising. Therefore, it is necessary to redefine the compensation voltage considering the slope of the output capacitor voltage, especially for the low current region where $T_{off}$ is not saturated.

The regions A, B and C are non-controllable voltages caused by the output capacitors in Figure 9, and require appropriate voltage compensation to get the ideal inverter output. The voltage region $\Delta v_{C_2}$ made by the output capacitor can be described as

$$\Delta v_{C_2} = A - (B + C),\tag{14}$$

$$A = \frac{1}{2}\left(\frac{V_{dc}}{2}\frac{T_{off}}{2T_s}\right),\tag{15}$$

$$B = \frac{1}{2}\left(-\frac{V_{dc}}{2}\frac{T_{off}}{2T_s}\right),\tag{16}$$

$$C = -\frac{V_{dc}}{2}\frac{T_{off}}{2T_s},\tag{17}$$

$$\Delta v_{C_2} = \frac{T_{off}}{2T_s}V_{dc}.\tag{18}$$

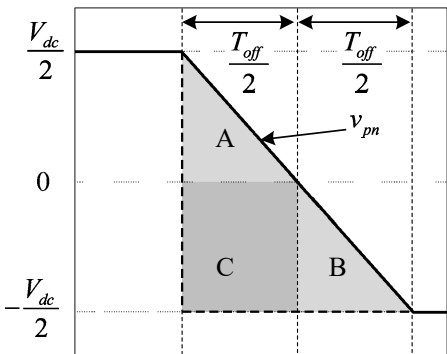

**Figure 9.** Detail of the dashed box in Figure 5c.

Equation (2) can be redefined as Equation (20) in order to properly compensate the repercussions of the output capacitor in the low-power area, in which the turn off delay of the switch has the greatest influence on the inverter output.

$$\therefore V_d = \frac{T_{on} + T_d - \frac{T_{off}}{2}}{T_s} V_{dc} \quad \left(T_d \geq T_{off}\right) \cdot \tag{19}$$

## 3. The Proposed DTCS

As mentioned above, the voltage error not only caused by dead-time distortion, but also caused by switch parasitic components, should be compensated to obtain the ideal three-phase VSI output. In Equation (19), generally the dead-time $T_d$ is fixed value, but the delay time $T_{off}$ is not. Hence, the precise $T_{off}$ has to be calculated according to the phase current levels in real-time for correct compensation. In this paper, to simplify the variation of $T_{off}$, TCV is adopted [8,15]. In addition, the novel three-phase TCV implementation strategy is proposed to simplify the realized trapezoidal shape, and the novel on-line TCV controller is also proposed to flexibly respond with the variation of the parameters.

### 3.1. Implementation of the TCV Based on the Current Position

The proposed DTCS uses a synchronous reference frame transformation matrix and limiter function to simplify realizing the TCV. Figure 10 displays the triangle waveform function $f(t)$, the sinusoidal waveform function $g(t)$ with peak value $k$ and the trapezoidal waveforms to compare the waveform outlines. In Figure 3, the position $\theta_d$ of the three-phase current can be calculated as follows using the *d-q* axis currents.

$$\delta = \tan^{-1}\left(\frac{i_d}{i_q}\right), \tag{20}$$

$$\theta_d = (\theta_e - \delta). \tag{21}$$

The three-phase sinusoidal waveforms, which is in phase with the three-phase current vector $I_s$, can be defined as follows. The $\alpha - \beta$ axis voltage $g(\Delta v_\alpha), g(\Delta v_\beta)$ with peak value $k$ on the stationary reference frame expressed as:

$$\begin{aligned} g(\Delta v_\alpha) &= -k \sin \theta_d \\ g(\Delta v_\beta) &= k \cos \theta_d \end{aligned} \tag{22}$$

$g(\Delta v_\alpha), g(\Delta v_\beta)$ is transferred to a three-phase stationary coordinate and the amplitude is limited to $\pm V_d$, as in Equation (23), to generate the TCV as shown in Figure 10b.

$$
\begin{cases}
g(\Delta v_{an}) = g(\Delta v_\alpha) & (-V_d \le g(\Delta v_{an}) \le V_d) \\
g(\Delta v_{bn}) = -\frac{(g(\Delta v_\alpha) - \sqrt{3}g(\Delta v_\beta))}{2} & (-V_d \le g(\Delta v_{bn}) \le V_d) \\
g(\Delta v_{cn}) = -\frac{(g(\Delta v_\alpha) + \sqrt{3}g(\Delta v_\beta))}{2} & (-V_d \le g(\Delta v_{cn}) \le V_d)
\end{cases}
\tag{23}
$$

As can be seen in the Figure 10b, if the peak level $k$ is large enough to approximate a linear slope between $V_d$ and $-V_d$, then the waveforms can be reckoned as $f(\Delta v_{an}) \approx g(\Delta v_{an})$. Next, as illustrated in Figure 11, the peak value $k$ for implementing the TCV having slopes of the width $\phi$ can be defined as follows.

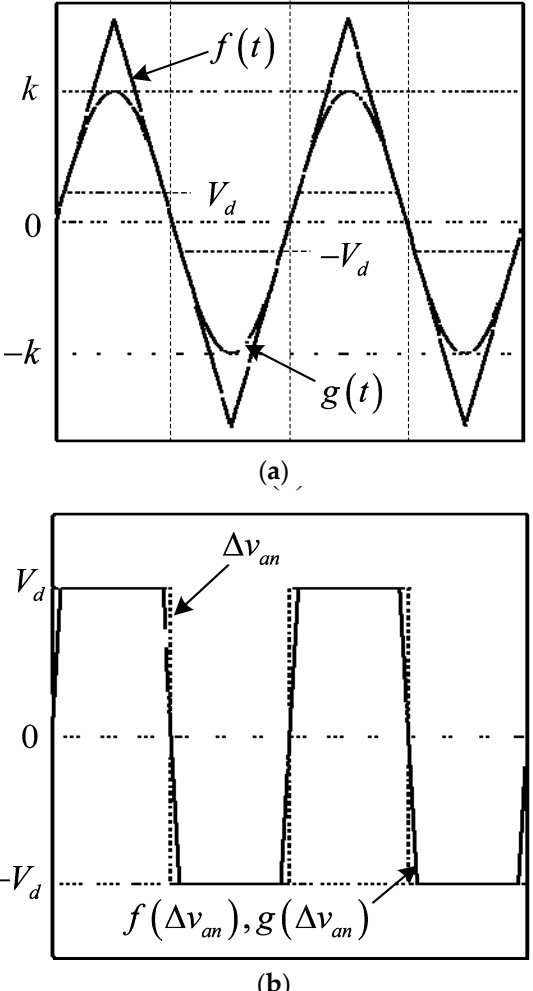

**(a)**

**(b)**

**Figure 10.** Proposed TCV implementation strategy; (**a**) triangle waveform function $f(t)$ and sinusoidal waveform function $g(t)$ with peak $k$; (**b**) TCV shapes comparison.

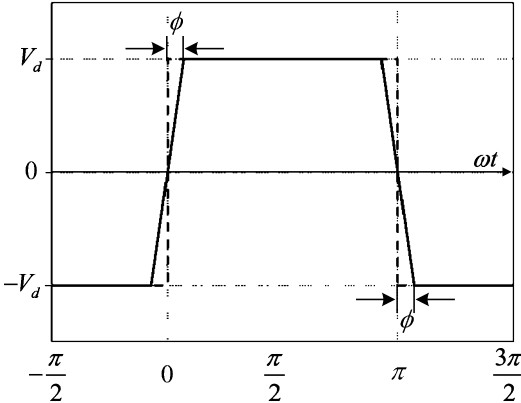

**Figure 11.** TCV with slope of the width $\phi$.

The function $g(t)$ can be expressed as $g(t) = k\sin(\omega t)$, and at the point $t_\phi$ when $g(t)$ has a slopes of the width $\phi$ defines as $t_\phi$, then the time $t_\phi$ can be derived as:

$$t_\phi = \frac{\phi}{\omega}. \tag{24}$$

Assuming that the output of $g(t)$ is $|V_d|$ at the time $t_\phi$, the $g(t)$ rewritten as:

$$k\,\sin\!\left(\omega t_\phi\right) = |V_d|, \tag{25}$$

$$\therefore k = \frac{|V_d|}{\sin(\phi)}. \tag{26}$$

The adjustable TCV having a desired compensation voltage amplitude $|V_d|$ and the compensation voltage slopes of the width $\phi$ can be realized with Equations (23) to (26).

### 3.2. The On-Line TCV Controller

It can be seen from Figure 6 and Equation (19) that the amplitudes and slopes of APVEs are changing according to the amount of the current flowing in the phase. Especially, when the VSI operates in the low current region, the magnitudes of the APVEs are decreased and the slopes of the TCV are increased. For these reasons, for smooth dead-time compensation in wide current regions, both the scale of the APVE and the slope of the TCV must be modulated to the optimum value corresponding to the inverter operating environment.

Using the previously defined Equations (12), (19) and (26), it might be possible to vary the amplitudes and the slopes of APVEs by responding to $T_{off}$. However, there are limitations to actively changing conditions. Therefore, the on-line TCV controller using the errors is proposed to implement robust dead-time compensation even at parameters with mismatching conditions.

While the influence of the dead-time appears $6n$th harmonics in the synchronous reference frame as Equation (4), the TCV errors can be obtained through them [16]. Although the $6n$th harmonics appear on both $d$-$q$ axis, the TCV error extracting axis can be selected as a $d$-axis since the $d$-axis has a larger voltage error than the $q$-axis. However, if there is a $d$-axis current ($\delta \neq 0$), the fundamental component of the harmonic voltages is moving to the $q$-axis as shown in Equation (4). Thus, to obtain a constant error regardless of the amount of $d$-axis current, the synchronous reference frame transformation based the three-phase current vector $I_s$ should be carried out. In this case, if the phase of the three-phase current vector $I_s$ is simply obtained by using the Equations (21) and (22), the harmonic components do not appear in the $d$-axis current because the harmonic component of the current affects $\delta$. For this reason, the ideal phase of three-phase current vector $I_s$ can be obained by $d$-$q$ axis current

commands $i_d^*, i_q^*$ as following Equations (27) and (28), assuming that the actual currents do not deviate for the current commands.

$$\delta^* = \tan^{-1}\left(\frac{i_d^*}{i_q^*}\right),$$ (27)

$$\theta_d^* = (\theta_e - \delta^*).$$ (28)

Figure 12 demonstrates the proposed on-line TCV controller scheme. To adjust the turn off delay time and slopes of the TCV, the integrators are designed, and for faster dynamics, $T_{ff}$ (12) is feedforwarded. The controller error $i_{dd}$ of Figure 12 is calculated from the d-q axis transformation matrix Equation (4) and the phase of ideal three-phase current vector $I_s$ Equation (28).

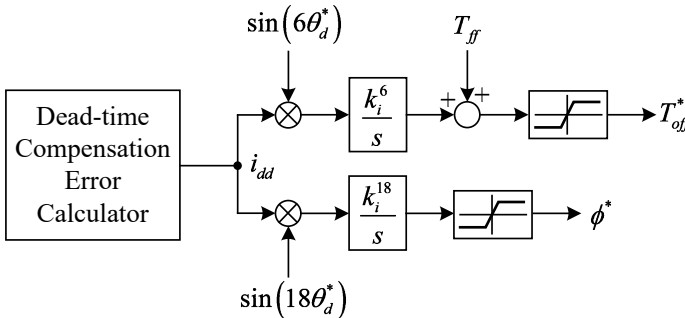

**Figure 12.** Proposed on-line TCV controller scheme.

The d-axis current (controller error) $i_{dd}$ based on $\theta_d^*$ includes the current distortion of the 6nth harmonic according to the influence of the dead time, and the polarity of the compensation voltage error can be determined by multiplying each harmonic order implemented using $\theta_d^*$ [16]. As the $T_{off}^*$ is a factor controlling the maximum value of APVE, the phase current $|i_p|$ in Equation (12) must be altered as the magnitude of the three-phase current vector $I_s$.

Figure 13 indicates the fast Fourier transform (FFT) results for two types of the DTCV errors. Figure 13a shows the voltage waveform of the $V_d$ amplitude error. Additionally, the FFT result of the $V_d$ error has a prominent component in the 6th harmonic as seen in Figure 13b. Figure 13c shows the voltage waveform of the slopes error of TCV. In addition, the FFT result of the slopes error has noticeable elements in the 12th and 18th harmonics, as seen in Figure 13d.

By utilizing the results of the Figure 13, the 6th and 18th can be selected as the multiplying frequency of $T_{off}$ and $\phi$, respectively, such as in Figure 12. In fact, the multiplying frequency for $\phi$ can be selected for both the 12th and the 18th as shown in Figure 13d. However, to minimize the influence of the $T_{off}$ error, the 18th is chosen instead of the 12th.

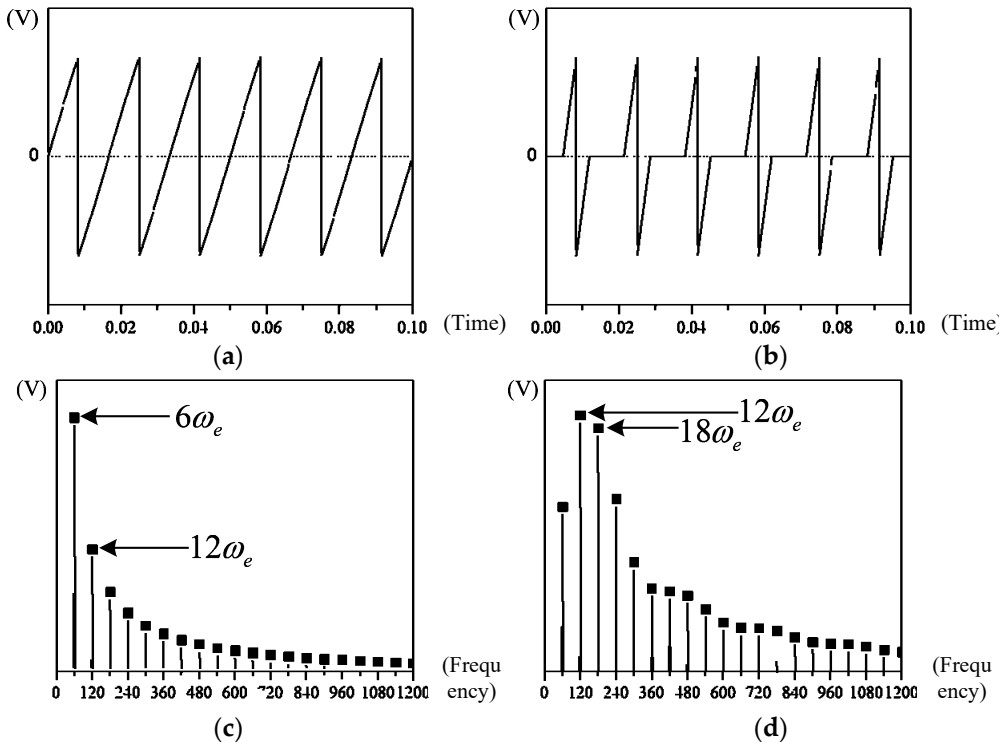

**Figure 13.** Analysis of DTCV error characteristics on d-axis; (**a**) error voltage waveform of $V_d$; (**b**) fast Fourier transform (FFT) result of Figure 13a; (**c**) error voltage waveform of slopes of DTCV; (**d**) FFT result of Figure 13c.

## 4. The Analysis of the Linear Modulation Region of the Three-Phase VSI with Proposed DTCS

The proposed DTCS is the method feedforwarding DTCV at the controller output, which is the voltage references. Thus, when the correct DTCV is applied, the controller side can consider that the inverter is ideal. However, if the controller outputs a voltage command exceeding the inverter output capable, the feedforwarded compensation voltage will not be able to suitably compensate due to physical constraints of the hardware. Therefore, it is required to limit the voltage reference of the controller by applying the proper physical voltage limit to the controller in order to perform normal operating of the DTCS.

### 4.1. Definition of the MMPV of the Three-Phase VSI

Table 1 and Figure 14 express the voltage vectors of the three-phase VSI in Figure 1. The hexagonal region $\mathfrak{R}^i$ using the six active voltage vectors is the ideal voltage region. The switching operation state function $Q(Q_n, Q_n, Q_n)$ of each leg in Figure 14 is shown as

$$Q(Q_1, Q_2, Q_3) \begin{cases} Q_n = 1: & Q_n = on, \overline{Q}_n = off \\ Q_n = 0: & \overline{Q}_n = on, Q_n = off \end{cases}. \tag{29}$$

In Figure 14, $V_{smax}^i$ is the magnitude of the MMPV in the ideal voltage region $\mathfrak{R}^i$. When an arbitrary voltage reference $V^*$ exists at $0° \le \theta \le 60°$, it can configure with the neighboring active voltage vector $V_1^i, V_2^i$ and zero voltage vector $O$ during the switching period $T_s$.

$$\int_0^{T_s} V^* dt = \int_0^{T_1} V_1^i dt + \int_{T_1}^{T_1+T_2} V_2^i dt + \int_{T_1+T_2}^{T_s} O dt, \tag{30}$$

where $T_1$, $T_2$ represent the interval for which the vectors $V_1^i$, $V_2^i$ is applied, respectively. The maximum active voltage vector with $V_1^i$, $V_2^i$ is

$$V^* T_s = V_1^i T_1 + V_2^i T_2. \tag{31}$$

The reference vector $V^*$ projected on the $V_1^i$ and $V_2^i$ vectors, respectively, can be given as

$$V_1^i T_1 = V^* T_s \cos\theta - V_2^i T_2 \cos 60°, \tag{32}$$

$$V_2^i T_s = V^* T_s \frac{\sin\theta}{\cos 30°}, \tag{33}$$

$$T_1 = \gamma T_s \cos\theta - \frac{\gamma T_s}{\sqrt{3}} \sin\theta \quad \left( \text{ where } \quad \gamma = \frac{3}{2}\frac{V^*}{V_{dc}} \right), \tag{34}$$

$$T_2 = \frac{2\gamma T_s}{\sqrt{3}} \sin\theta. \tag{35}$$

Here, since $T_1 + T_2 \leq T_s$, Equations (34) and (35) can be derived as Equation (36).

$$V^*\left(\cos\theta + \frac{1}{\sqrt{3}}\sin\theta\right) \leq \frac{2}{3}V_{dc}, \tag{36}$$

$$V^* \leq \frac{V_{dc}}{\sqrt{3}}\frac{1}{\sin(\theta + 60°)}, \tag{37}$$

$$\therefore V_{smax}^i = \frac{V_{dc}}{\sqrt{3}} \quad (\text{where} \quad \theta = 30°). \tag{38}$$

Accordingly, the MMPV amplitude $V_{smax}^i$ at $\theta = 30°$ in the ideal three-phase VSI can be defined as Equation (38).

**Table 1.** The phase voltages and space voltage vectors of the typical three-phase VSI.

| Vector | Phase Voltage | | | Space Voltage Vector |
|--------|---------------|---|---|----------------------|
| | $v_{as}$ | $v_{bs}$ | $v_{cs}$ | |
| $V_1^i$ | $\frac{2}{3}V_{dc}$ | $-\frac{1}{3}V_{dc}$ | $-\frac{1}{3}V_{dc}$ | $\frac{2}{3}V_{dc}/0°$ |
| $V_2^i$ | $\frac{1}{3}V_{dc}$ | $\frac{1}{3}V_{dc}$ | $-\frac{2}{3}V_{dc}$ | $\frac{2}{3}V_{dc}/60°$ |
| $V_3^i$ | $-\frac{1}{3}V_{dc}$ | $\frac{2}{3}V_{dc}$ | $-\frac{1}{3}V_{dc}$ | $\frac{2}{3}V_{dc}/120°$ |
| $V_3^i$ | $-\frac{2}{3}V_{dc}$ | $\frac{1}{3}V_{dc}$ | $\frac{1}{3}V_{dc}$ | $\frac{2}{3}V_{dc}/180°$ |
| $V_5^i$ | $-\frac{1}{3}V_{dc}$ | $-\frac{1}{3}V_{dc}$ | $\frac{2}{3}V_{dc}$ | $\frac{2}{3}V_{dc}/240°$ |
| $V_6^i$ | $\frac{1}{3}V_{dc}$ | $-\frac{2}{3}V_{dc}$ | $\frac{1}{3}V_{dc}$ | $\frac{2}{3}V_{dc}/300°$ |
| $O$ | $0$ | $0$ | $0$ | $0/0°$ |

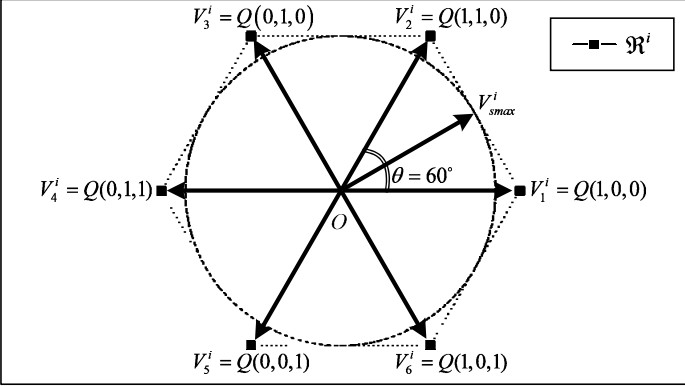

**Figure 14.** Six output voltage vectors of typical three-phase VSI.

### 4.2. Analysis of the Linear Modulation Region with Dead-Time and When the Proposed DTCS is Applied

The distortion voltage from the dead-time can be derived by using Equations (3) to (6) and it is illustrated in Figure 4. In this case, the affection of the distorted voltage caused by the dead-time depends on the phase $\psi$ of the current. Thus, the distortion voltage $\Delta v_\alpha^{err}(\psi)$, $\Delta v_\beta^{err}(\psi)$ can be defined as the function of $\psi$ on the $\alpha - \beta$ axis. Where the maximum three-phase VSI output with six active voltage vectors is $v_\alpha^i$, $v_\beta^i$, the distorted three-phase VSI output $v_\alpha^r$, $v_\beta^r$ can be expressed as

$$\begin{bmatrix} v_\alpha^r \\ v_\beta^r \end{bmatrix} = \begin{bmatrix} v_\alpha^i \\ v_\beta^i \end{bmatrix} + \begin{bmatrix} \Delta v_\alpha^{err}(\psi) \\ \Delta v_\beta^{err}(\psi) \end{bmatrix}. \tag{39}$$

The Figure 15 shows example of the phase $\psi$ between the voltage reference and phase current.

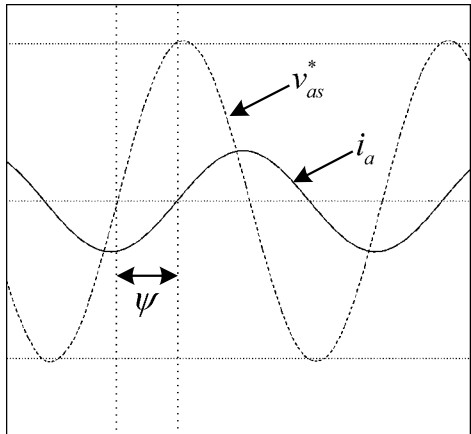

**Figure 15.** The current phase $\psi$ with a-phase voltage reference $v_{as}^*$ and a-phase current $i_a$.

Figure 16 illustrates the distorted three-phase VSI output voltage waveforms $v_\alpha^r$, $v_\beta^r$ on the stationary reference frame $\alpha - \beta$ axis according to the phase $\psi$. Figure 17 displays $v_\alpha^r$, $v_\beta^r$ regions on the x-y plot using the waveforms of Figure 16 and Equation (39), and the right side of each voltage region reveals the sector ① in detail for more accurate analysis.

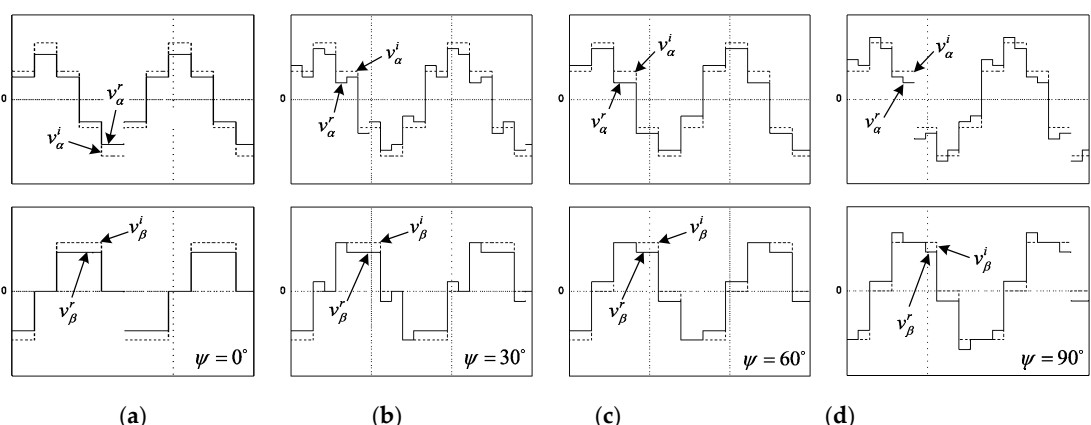

| (a) | (b) | (c) | (d) |

**Figure 16.** $v_\alpha^i$, $v_\beta^i$ and $v_\alpha^r$, $v_\beta^r$ waveforms on the stationary reference frame according to the phase $\psi$; (**a**) $\psi = 0°$; (**b**) $\psi = 30°$; (**c**) $\psi = 60°$; (**d**) $\psi = 90°$.

Figure 17a–d was divided into four regions along the voltage region forms. In the case of $\psi$ being a negative phase value (leading condition), it has the same form with a positive phase value as Figure 17, since the distortion voltage is an even function. Therefore, the MMPV magnitudes are arranged in Table 2 instead of illustrating the regions about the negative phase value. $\mathfrak{R}^i$, $\mathfrak{R}^r$ and

$\mathfrak{R}^c$ are the output voltage region of the ideal three-phase VSI, the output voltage of the three-phase VSI distorted by the dead-time, and the output voltage region of the three-phase VSI applied to the proposed DTCS, respectively. As the dead-time physically limits the turn on period of the switch, even if the dead-time compensated theoretically, the physical limits of the inverter cannot be recompensed. Consequently, when the proposed DTCS is applied, the compensated voltage region $\mathfrak{R}^c$ is inscribed within the dead-time voltage region $\mathfrak{R}^r$.

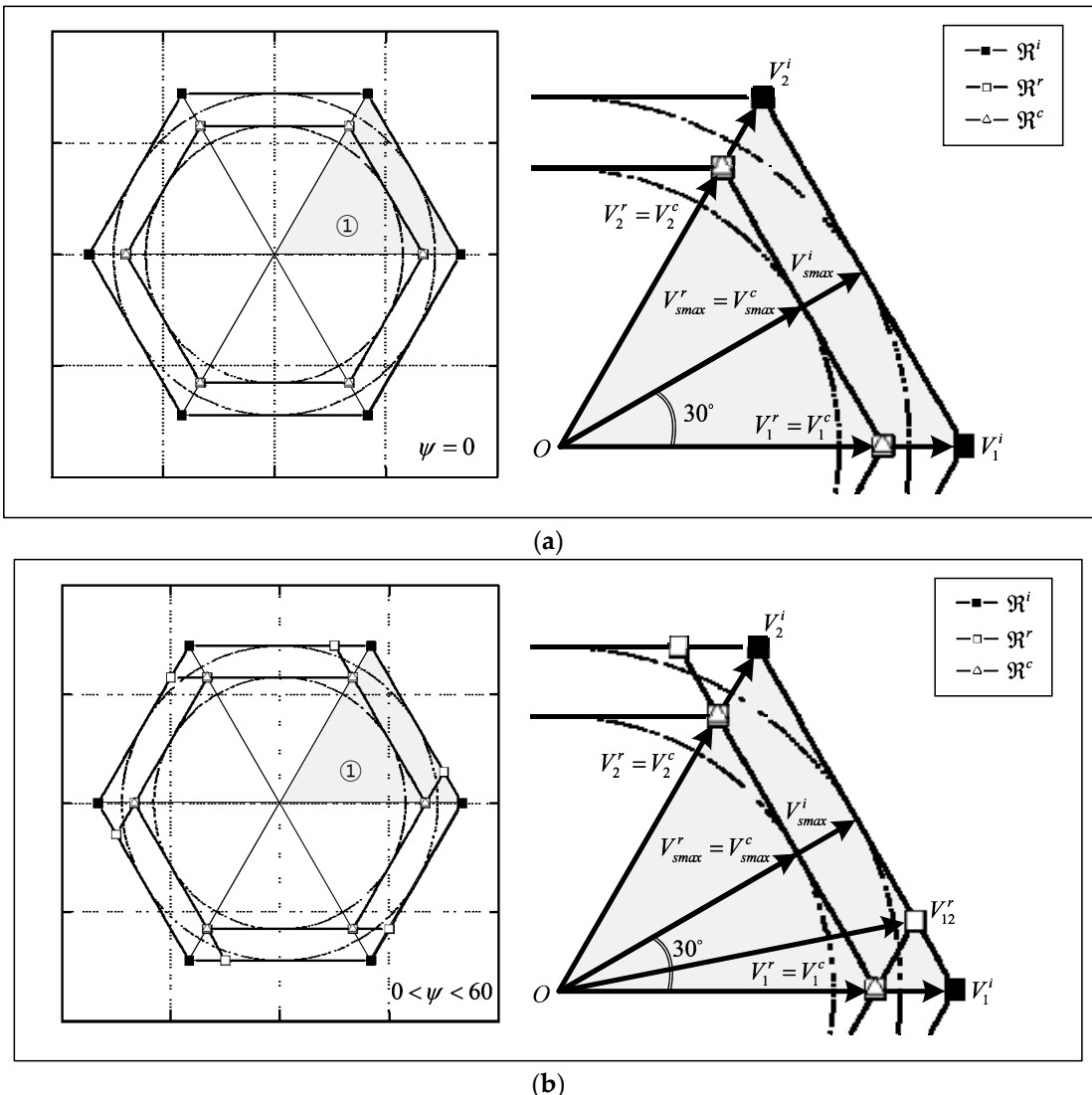

(**a**)

(**b**)

**Figure 17.** *Cont.*

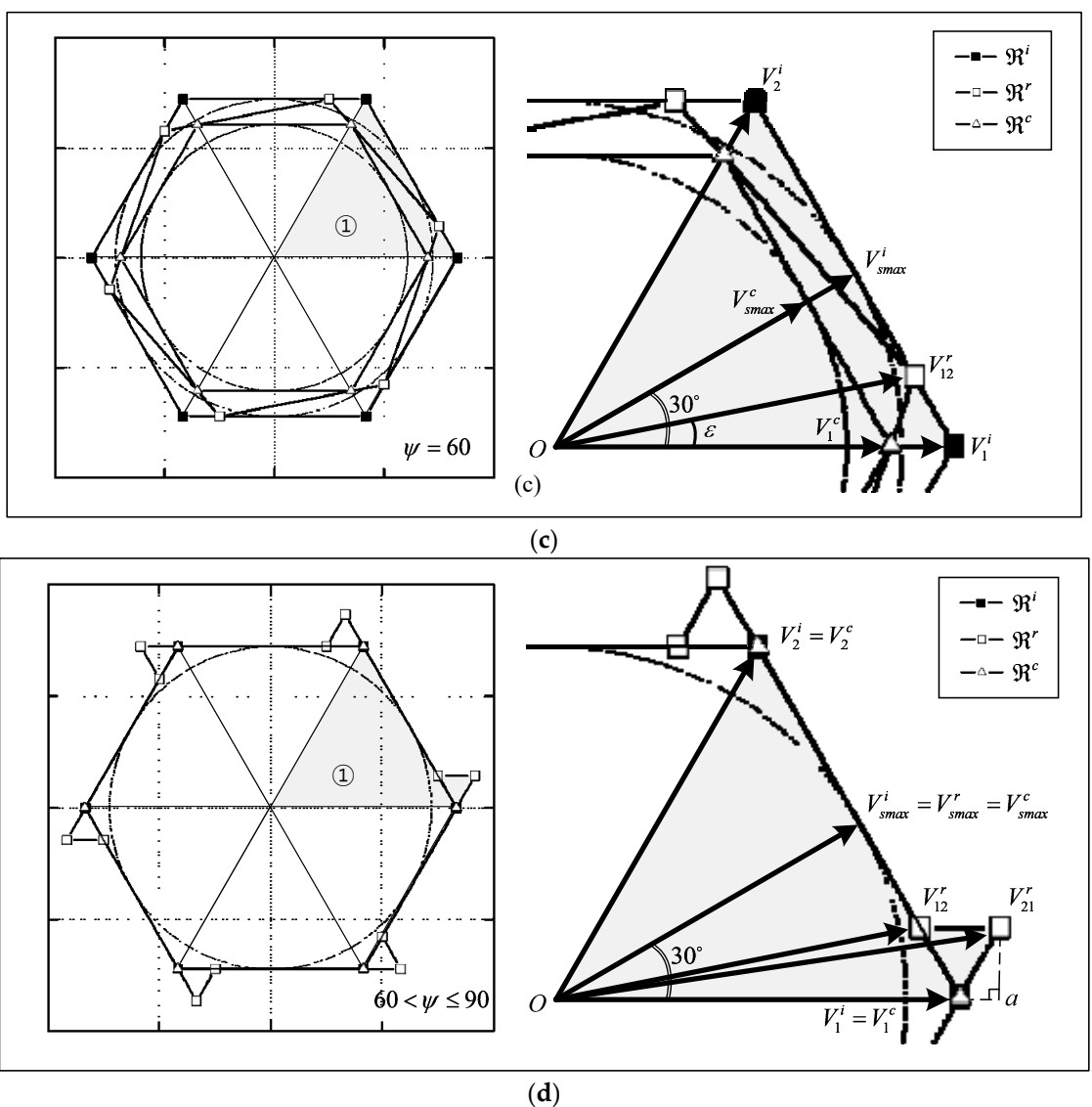

**Figure 17.** Distorted voltage regions and voltage vectors according to the current phase $\psi$; (**a**) $\psi = 0°$; (**b**) $0° < \psi < 60°$; (**c**) $\psi = 60°$; (**d**) $60° < \psi \leq 90°$.

**Table 2.** The compensated MMPV $V_{smax}^c$ when the proposed DTCS applied.

| Phase Delay of the Current | MMPV $V_{smax}^c$ Magnitude |
|:---:|:---:|
| $\psi = 0°$ | from Equation (42) $\frac{V_{dc} - 2V_d}{\sqrt{3}}$ |
| $0° < \psi < 60°, -60° < \psi < 0°$ | from Equation (43) $\frac{V_{dc} - 2V_d}{\sqrt{3}}$ |
| $\psi = 60°, \psi = -60°$ | from Equation (46) $\frac{\sqrt{3}}{2}\left|V_1^c\right|$ |
| $60° < \psi \leq 90°, -90° \leq \psi < -60°$ | from Equation (50) $\frac{V_{dc}}{\sqrt{3}}$ |

### 4.2.1. Where $\psi = 0°$

When the output voltage is in phase with phase current, the voltage distortion exactly coincides with the six active voltage vectors as in Figure 16a. Hence, by using the above Equations (31) to (39), the arbitrary voltage reference $V^*$ in the sector ① can be expressed as follows using the neighboring real voltage vectors $V_1^r, V_2^r$.

$$\begin{aligned}
\left|V_1^r\right| &= \left|V_1^c\right| = \left|V_1^i\right| - \frac{4}{3}V_d \\
\left|V_2^r\right| &= \left|V_2^c\right| = \left|V_2^i\right| - \frac{4}{3}V_d
\end{aligned}, \tag{40}$$

$$V^*\left(\cos\theta + \frac{1}{\sqrt{3}}\sin\theta\right) \leq \frac{2}{3}(V_{dc} - 2V_d), \tag{41}$$

$$\therefore V^r_{smax} = \frac{V_{dc}-2V_d}{\sqrt{3}} \quad (where \quad \theta = 30°) , \tag{42}$$

$$\therefore V^r_{smax} = V^c_{smax} = \frac{V_{dc}-2V_d}{\sqrt{3}} \quad (where \quad \theta = 30°) . \tag{43}$$

### 4.2.2. Where $0° < \psi < 60°$

The proportions of the voltage vectors in Figures 17b and 16b can be expressed as:

$$V^r_{12} = \frac{2}{3}(V_{dc} - V_d) + j\frac{2}{3\sqrt{3}}V_d, \tag{44}$$

$$|V^r_1| = |V^c_1| = \frac{2}{3}(V_{dc} - 2V_d). \tag{45}$$

Since the distorted real voltage vector $V^r_{12}$ does not affect to the active voltage vector and output voltage region in Figure 17b, $V^c_{smax}$ can be derived as:

$$\therefore V^r_{smax} = V^c_{smax} = \frac{V_{dc}-2V_d}{\sqrt{3}} \quad (where \quad \theta = 30°) . \tag{46}$$

### 4.2.3. Where $\psi = 60°$

In Figure 17c, the distorted voltage vector $V^r_{12}$ affects the real output voltage region. It can be expressed as following, using Figure 16c:

$$V^r_{12} = \frac{2}{3}(V_{dc} - V_d) + j\frac{2}{\sqrt{3}}V_d. \tag{47}$$

In addition, the angle $\varepsilon$ between $V^r_{12}$ and $V^i_1$ is:

$$\varepsilon = \tan^{-1}\left(\frac{\frac{2}{\sqrt{3}}V_d}{\frac{2}{3}(V_{dc} - V_d)}\right), \tag{48}$$

$$|V^c_1| = \frac{2}{3}(V_{dc} - V_d) - \frac{2}{\sqrt{3}}V_d\frac{1}{\tan\left(\frac{\pi}{3} + \varepsilon\right)}, \tag{49}$$

$$\therefore V^c_{smax} = \frac{\sqrt{3}}{2}|V^c_1|. \tag{50}$$

### 4.2.4. Where $60° < \psi \leq 90°$

In Figure 17d, the voltage vectors $V^r_{12}$, $V^r_{21}$ that have arisen with dead-time can be expressed as follows, as in Figure 16d:

$$V^r_{12} = \frac{2}{3}(V_{dc} - V_d) + j\frac{2}{\sqrt{3}}V_d, \tag{51}$$

$$V^r_{21} = \frac{2}{3}(V_{dc} + V_d) + j\frac{2}{\sqrt{3}}V_d. \tag{52}$$

The degree $\angle V^r_{21}V^c_1 a$ is always $60°$ according to the Equation (53):

$$\tan^{-1}\left(\frac{2/\sqrt{3}}{2/3}\right) = 60°. \tag{53}$$

As the segment $\overline{V^r_{12}V^r_{21}}$ is parallel to the voltage vector $V^i_1$, the additional voltage region $\Delta V^r_{12}V^r_{21}V^i_1$ generated by the dead-time forms a regular triangle so that the voltage vector $V^r_{12}$ is always adjoined with ideal voltage region $\mathfrak{R}^i$. Therefore, the compensated MMPV vector $V^c_{smax}$ has equal magnitude with ideal modulation phase voltage vector $V^i_{smax}$.

$$\therefore V^c_{smax} = V^i_{smax} = \frac{V_{dc}}{\sqrt{3}}. \tag{54}$$

## 5. The Results of the Simulation and Experiment of the Proposed DTCS

The Figure 18 is a simplified block diagram of three-phase VSI controller including the proposed DTCS. Since the DTCV is feedforwarded at the controller output, there is no need to compensate for the dead-time into the current controller. Therefore, the error between the voltage reference of the current controller and the output voltage of the three-phase VSI can be minimized, and it makes it easy to design the algorithms using the voltage reference $v^*_{dL}, v^*_{qL}$. As mentioned above, unless the output of the current controller is appropriately limited, normal dead-time compensation is not possible, so the current controller output $v^*_d, v^*_q$ should be restricted as shown in block (b) in Figure 18. Here, the voltage limit can be defined according to the phase of the current in Table 2. The outside of the current controller of Figure 18 can be designed along the employed applications.

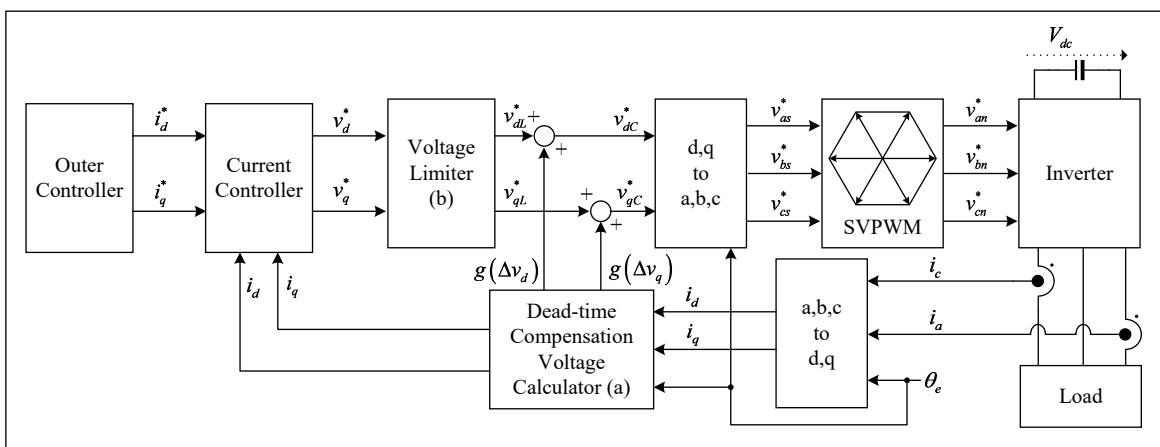

**Figure 18.** The control block diagram of three-phase VSI with proposed DTCS.

Figure 19 shows a detailed block diagram of the proposed DTCS of Figure 18a. The TCV contains the position calculating block which calculates and outputs current angles $\theta_d, \theta^*_d$ and the on-line TCV controller block which regulates $\phi$, $T_{off}$ and outputs the references $\phi^*$, $T^*_{off}$ and compensation voltage calculator block which realizes TCV.

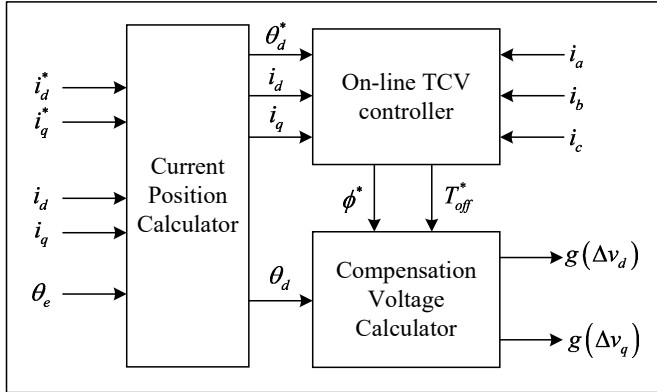

**Figure 19.** The specific block diagram of Figure 18a.

### 5.1. Simulation Results

The proposed DTCS was verified using the simulation software Psim. The three-phase VSI and DTCS were designed the same as in Figure 18, and the current controller was performed alone without outer control loop. The circuit uses a three-phase VSI as shown in Figure 1. In order to maximize the effect of dead-time, the load was composed of only the inductors and the resistors without the back electromotive force or the voltage sources. The switches modeled in SKM50GB063D manufactured by the SEMIKRON were used to observe the effects of the output capacitors into the simulation result. Detailed simulation specifications are shown in Table 3.

**Table 3.** The specifications of the simulation.

| Parameters | Description | Value | Parameters | Description | Value |
| --- | --- | --- | --- | --- | --- |
| $V_{dc}$ | Dc-link voltage level | 310 V | $V_{ce}$ | Maximum collector-emitter voltage rating | 600 V |
| $R_s$ | Phase resistance | 0.5 Ω | $v_{G_{th}}$ | Gate threshold voltage | 4.5 V |
| $L_s$ | Phase inductance | 10 mH | $t_f$ | Fall time of the current | 300 ns |
| $f_{sw}$ | Switching frequency | 10 kHz | $C_{ies}$ | Input capacitance | 2.2 nF |
| $T_d$ | Dead-time | 5.0 μs | $C_{oes}$ | Output capacitance | 2.2 nF |
| $f_m$ | Fundamental frequency | 50 Hz | $R_{ce\_on}$ | On resistance | 28 mΩ |

Figure 20 demonstrates the simulation results of the proposed DTCV with the above specifications. Figure 20a shows the compensated three-phase current waveforms and Figure 20d displays the compensation voltage on the synchronous reference frame *d-q* axis. It can be confirmed that the magnitudes of compensation voltages $g(\Delta v_d)$, $g(\Delta v_q)$ change according to the magnitude of the current vector $I_s$ and the fundamental component of the dead-time compensation voltage shifts to *d*-axis along *d*-axis current level. Figure 20e presents the position information of the three-phase current vector $I_s$ using the control position $\theta_e$ and Equation (21). It can validate that the position $\theta_d$ of the $I_s$ is changed along *d-q* axis current amounts.

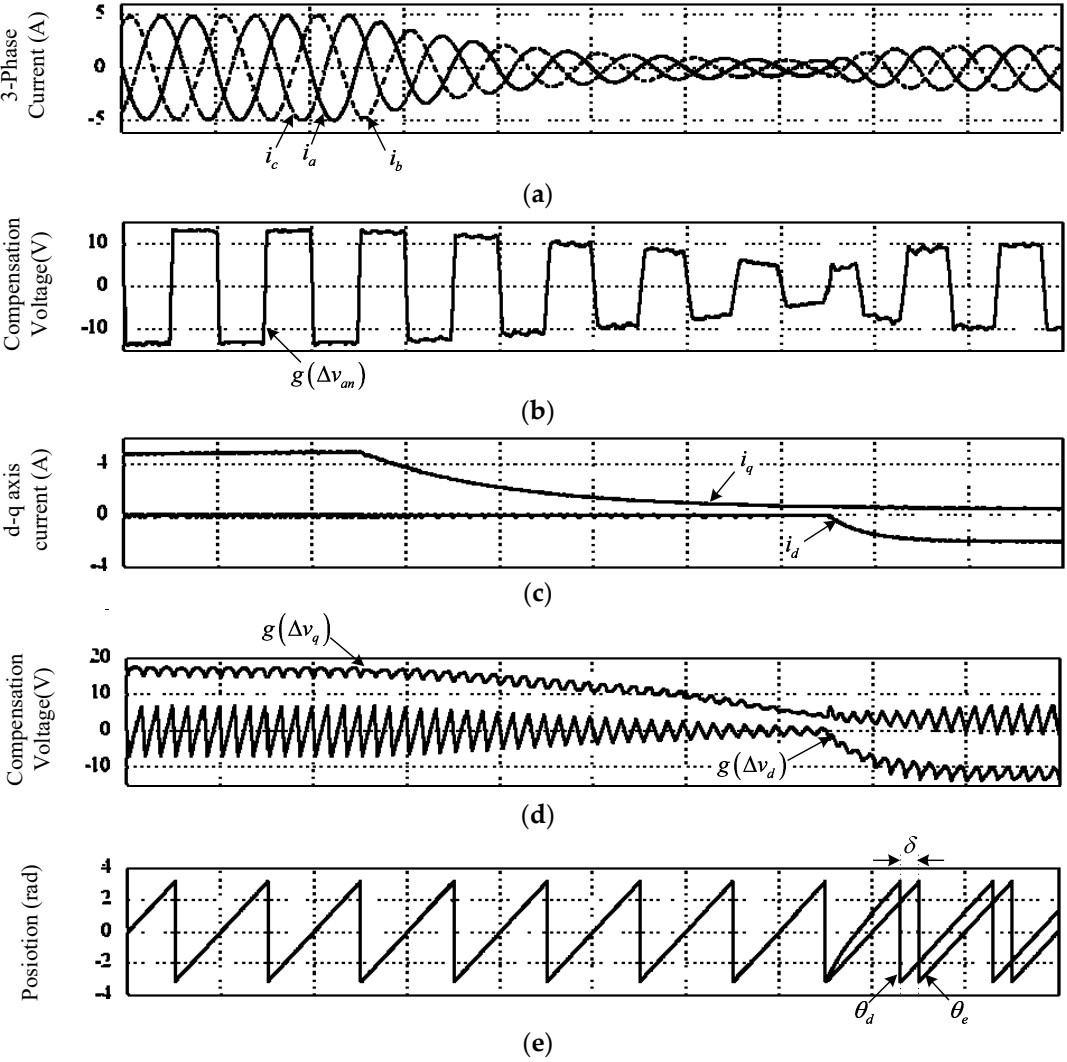

**Figure 20.** Simulation results of the proposed DTCS; (**a**) three-phase currents; (**b**) TCV of the a-phase; (**c**) *d-q* axis currents on the synchronous reference frame; (**d**) DTCV on the synchronous reference frame *d-q* axis; (**e**) positions.

Figure 21 indicates the simulation result when any DTCS is not applied under the condition of Figure 20, comparing the performance of the proposed DTCS. Figure 21a is the three-phase currents and Figure 21b is the *d-q* axis current on the synchronous reference frame. It can be seen the amount of the current decreases, and the harmonic distortions of the current become smaller. This is caused by the fact that as the switch turn off delay $T_{off}$ is increased in the low current region. However, when $T_{off}$ is in the saturation region sufficiently, there is notable current distortions because the current controller cannot compensate the voltage distortion of high order harmonic distortions. As a result, the proposed DTCS applied three-phase currents has THD below 0.4%. In contrast, the three-phase currents which are not applied DTCS have a 5.4% THD that is about 10 times larger.

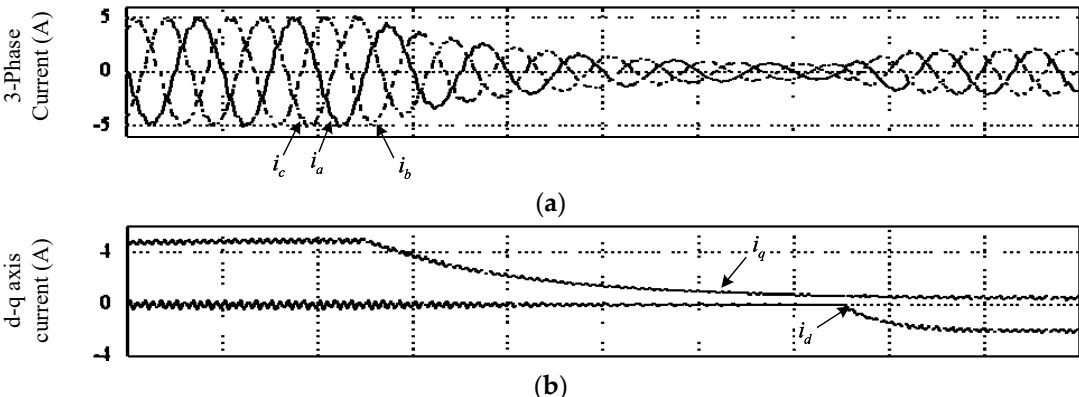

**Figure 21.** Simulation results without DTCS; (**a**) three-phase currents; (**b**) *d-q* axis currents on the synchronous reference frame.

*5.2. Experimental Results*

The experiment to verify the proposed DTCS used three-phase VSI connected with DC-power supply as shown Figure 22. To maximize the effects of dead-time, it applied the only inductors and resistances as a load. Detailed specifications of the experiment environment are summarized in Table 4.

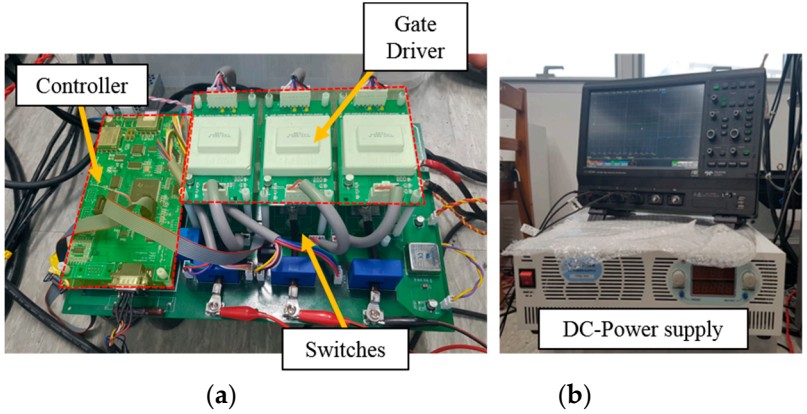

**Figure 22.** Experiment setting; (**a**) the three-phase VSI; (**b**) DC-power supply for the DC-link voltage source.

**Table 4.** Experiment specifications.

| Parameters | Description | Value |
|---|---|---|
| Switch ($Q_n$) | Three-phase VSI switch | SKM50GB063D |
| Gate driver | Gate driver of VSI | SKHI 22B |
| MCU | Micro controller unit | DSP 320F28335 |
| DC power supply | DC-link voltage source | TP5H-10D |
| $V_{dc}$ | DC-link voltage level | 310 V |
| $R_s$ | Phase resistance | 5.5 Ω |
| $L_s$ | Phase inductance | 20.5 mH |
| $f_{sw}$ | Switching Frequency | 10 kHz |
| $T_d$ | Dead-time | 5.0 μs |

Figures 23–25 show the three-phase currents waveforms and dead-time compensation pole voltage of the a-phase to compare the performance of the proposed DTCS. The three-phase current levels kept around 1.4% of the switch current rating to perform in the region where the effects of the switch parasitic are present. Figure 23 is the three-phase current waveforms without any DTCS, and it can be seen that serious current distortions occurs near the zero crossing and peak area. Figure 24 illustrates

the three-phase current waveforms when a conventional DTCS considering only dead-time $T_d$ is adapted. While the compensation voltage amplitudes that does not reflect the variation of $T_{off}$ are larger than the actual voltage errors. Hence, it confirmed that the current distortion is due to the excessive compensation voltage. Figure 25 reveals the currents waveforms when the proposed DTCS was applied, which shows very ideal sinusoidal current waveforms even in the low current region where affected by $T_{off}$ variation.

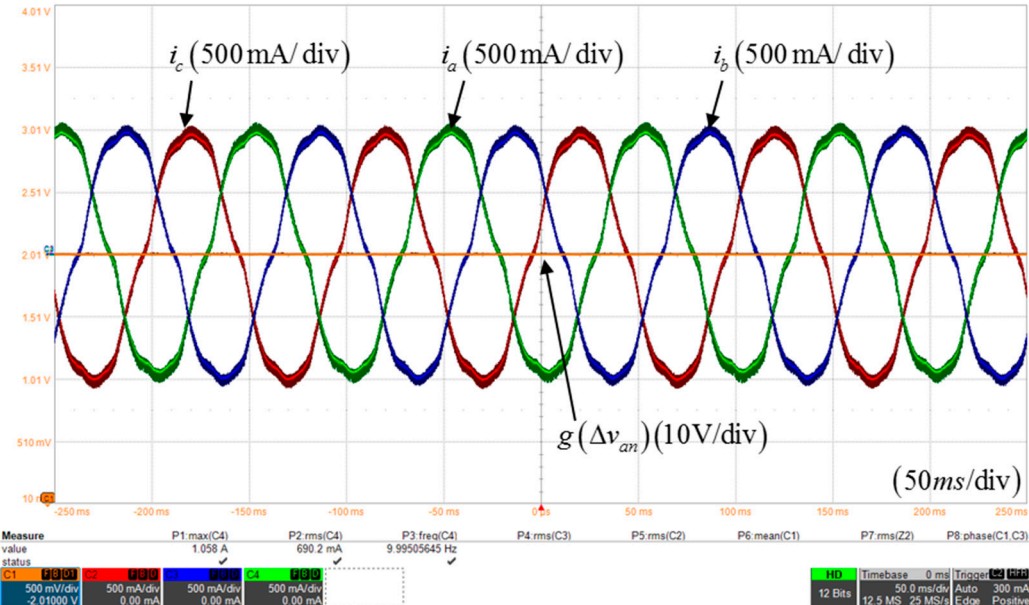

**Figure 23.** Three-phase currents waveforms and a-phase dead-time compensation pole voltage waveform without any DTCS.

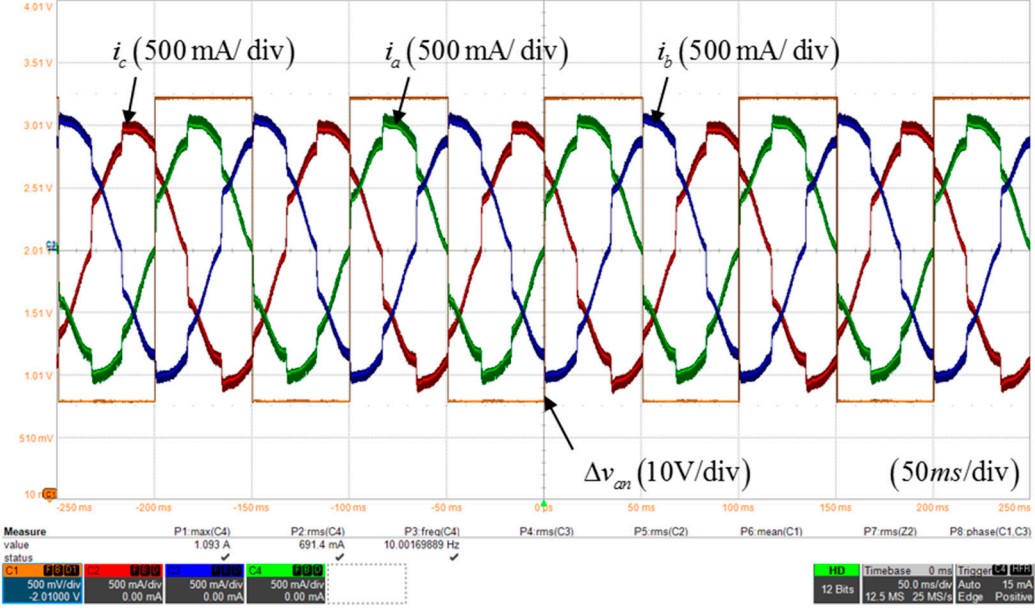

**Figure 24.** Three-phase currents waveforms and a-phase dead-time compensation pole voltage waveform when conventional DTCS considering only $T_d$ was applied.

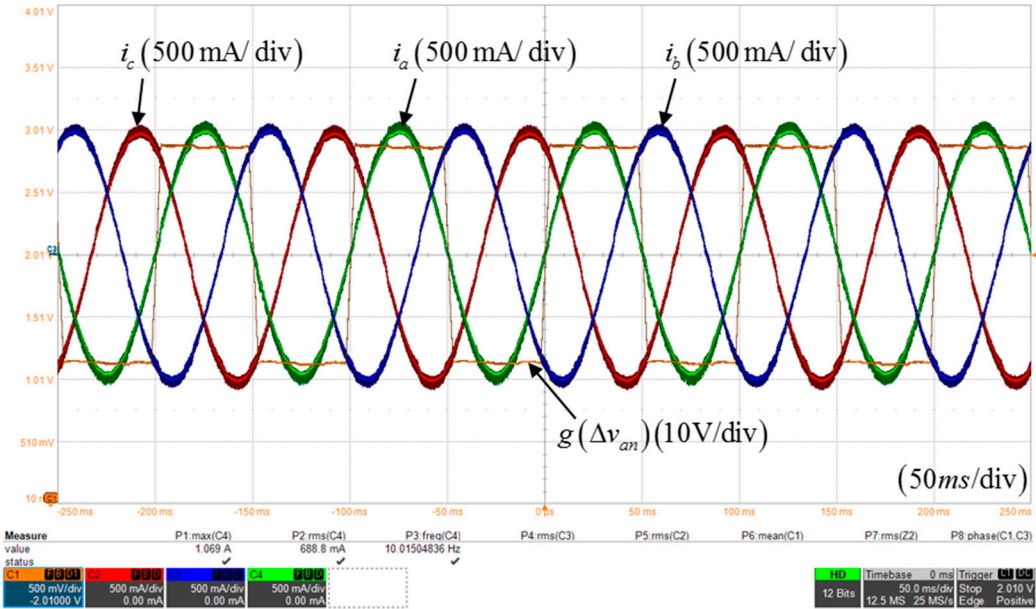

**Figure 25.** Three-phase currents waveforms and a-phase dead-time compensation pole voltage waveform when the proposed DTCS was applied.

The Figure 26 displays waveforms for confirming the effects of TCV. The compensating pole voltage in Figure 26 has a rectangle shape, whereas amplitude is equal with Figure 25. Even if it was compensating with proper compensation voltage level, the currents distortions still exist. These experiment results prove that not only the amplitudes but also the slopes of the compensation pole voltages are very important factors for correct dead-time compensation, especially in the low-current region.

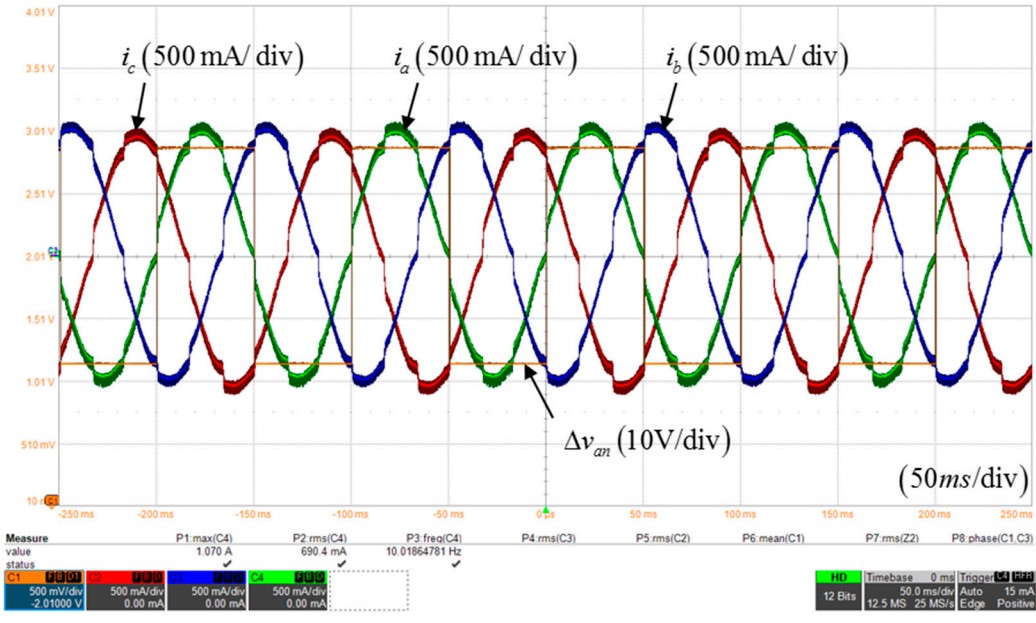

**Figure 26.** Three-phase currents waveforms and a-phase dead-time compensation pole voltage waveform when DTCS, considering only the proper voltage level, was applied.

The Figures 27–30 show $\alpha - \beta$ axis currents from above three-phase currents and also demonstrate the $\alpha - \beta$ axis currents on the x-y plot that can compare the distortion of the current more intuitively. The $\alpha - \beta$ axis current waveform on the x-y plot is closer to the ideal circle, and the more ideal currents.

The $\alpha - \beta$ axis currents were displayed using the DAC and since the $\alpha$-axis current is equal to the a-phase current, both currents waveforms were overlapped for check the function of DAC.

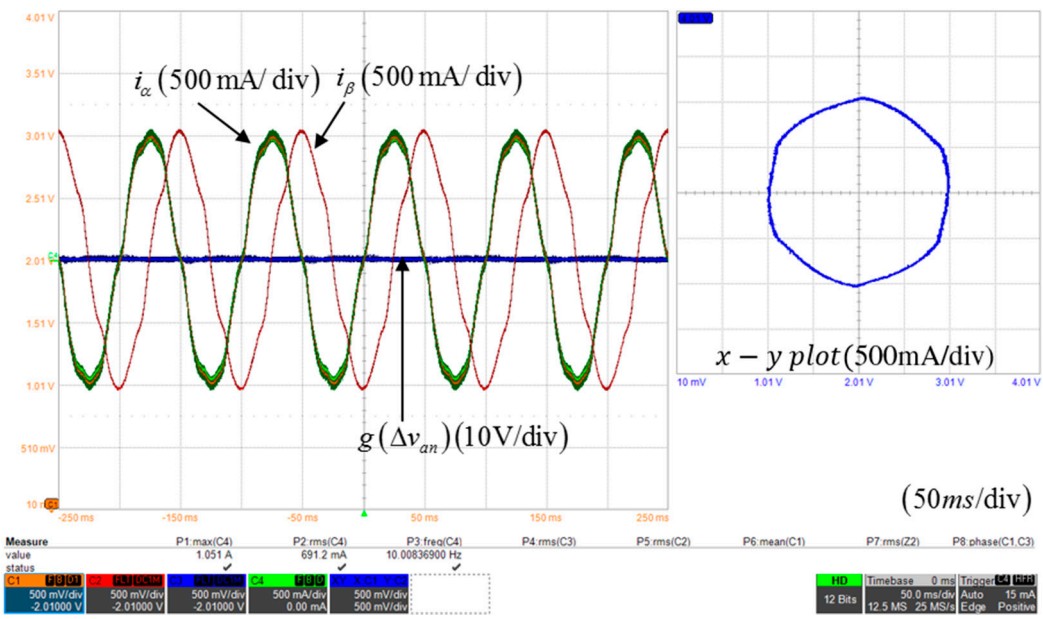

**Figure 27.** Stationary reference frame $\alpha - \beta$ axis and on the x-y plot current waveforms with Figure 23.

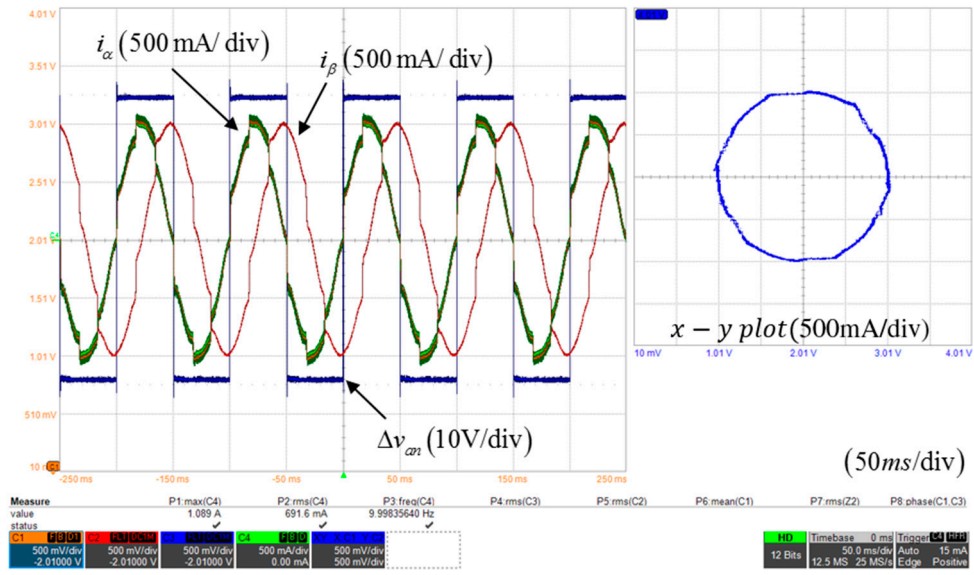

**Figure 28.** Stationary reference frame $\alpha - \beta$ axis and on the x-y plot current waveforms with Figure 24.

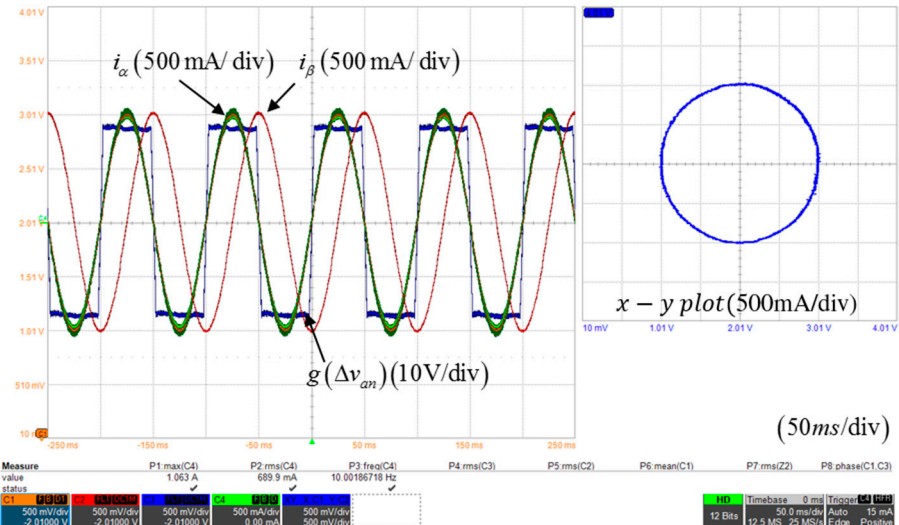

**Figure 29.** Stationary reference frame $\alpha - \beta$ axis and on the x-y plot current waveforms with Figure 25.

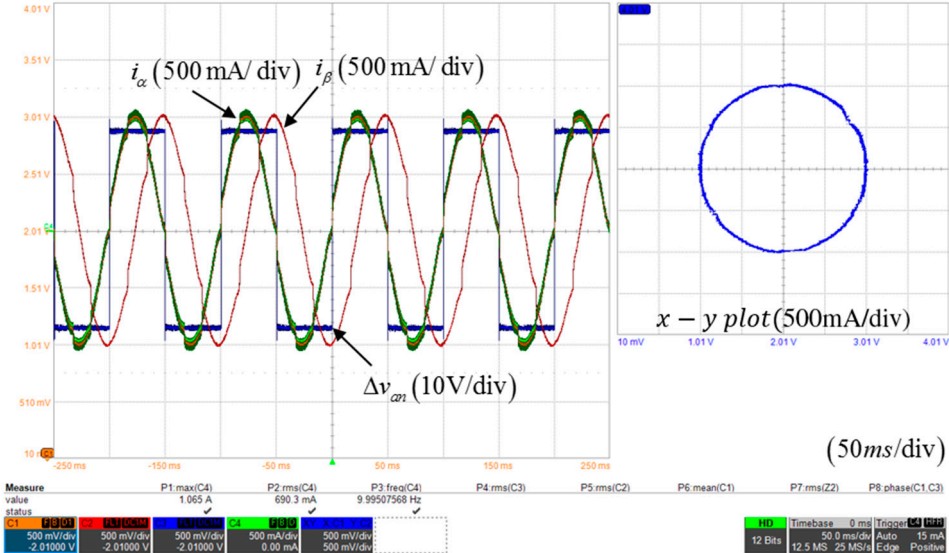

**Figure 30.** Stationary reference frame $\alpha - \beta$ axis and on the x-y plot current waveforms with Figure 26.

Figure 29 demonstrates the currents waveforms on the stationary reference frame when the proposed DTCS is applied. The currents waveform on the x-y plot is closer to the ideal circle than the waveforms in Figures 26 and 27. Additionally, the currents waveform in Figure 30 appear closer to the circle than in Figures 27 and 28 but it is impossible to draw the ideal circular waveform as proposed DTCS's.

Figure 31 shows the currents' THDs at various inverter output conditions to compare the performance of the proposed DTCS. The currents' THDs were compared with the magnitude of the peak current and the control frequency, and the THD of the a-phase current was extracted. In Figure 31, the square and circle symbols show the THD when the dead time compensation voltage is not applied and the THD when the compensation voltage considering only $T_d$ is applied as shown in Figures 23 and 24, respectively. The triangle symbol is the THD when proposed DTCS is applied. The vertical axis denotes the THD value of the a-phase current and the horizontal axis denotes the peak values of the three-phase current.

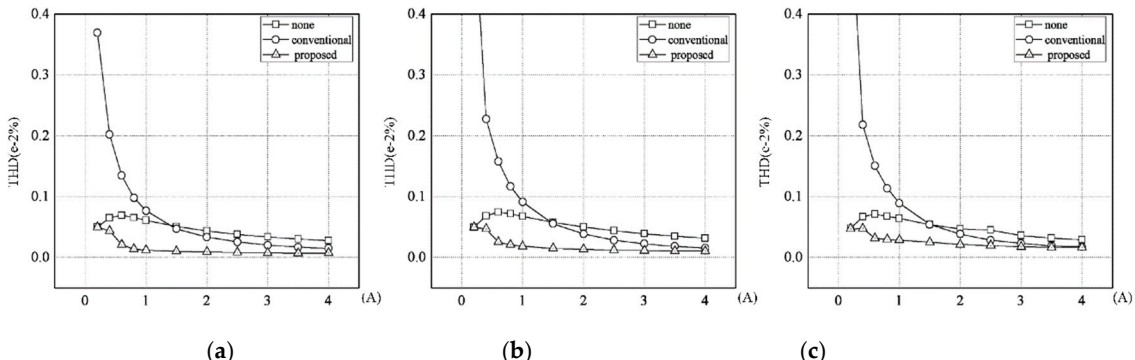

**Figure 31.** Comparing the a-phase current THD; (**a**) 10 Hz; (**b**) 30 Hz; (**c**) 50 Hz.

It can be seen that the current distortion due to the compensating voltage error becomes serious in the low power period where the influence of $T_{off}$ becomes very large when the compensation voltage considering only $T_d$ is applied, and the current THD gradually decreases as the peak current increases. These results show that the application of the wrong compensation voltage has a greater adverse effect than without. In the case where no DTCS is employed, the THD is low in the low current region since the influence of the dead-time is cancelled out due to the $T_{off}$. However, as the magnitude of the current rises, the influence of $T_{off}$ is reduced, so that the current distortion gradually appears. If the current magnitude increases until the duty ratio becomes relatively larger than the dead time ratio, the influence of the dead time is reduced and the current THD is lowered. The proposed DTCS shows much lower current THD in wide-current domains. In particular, applying the real-time adjusted TCV using the on-line TCV controller shows higher performance than a general compensation algorithm in a low-power section.

Figure 32 shows the current THD according to the current control frequency when the proposed DTCS is employed. In Figure 32, it can be seen that the current THD increases as the current control frequency increases, because it cannot synthesize the exact trapezoidal voltage waveform due to the limit of the fixed switching frequency. Therefore, with increased control frequency, the current THD increases due to the imperfect TCV.

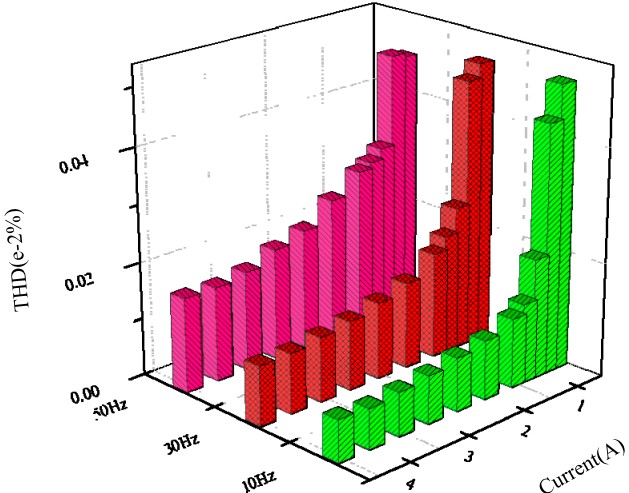

**Figure 32.** Changes of current THD according to frequency change of proposed DTCS.

## 6. Conclusions

In this paper, the analysis of the output voltage distortion of the three-phase VSI due to the dead-time and the output capacitor was carried out, and a novel DTCS for compensating the nonlinearly varying voltage distortion was proposed. First, equations and strategies for simplified

TCV implementation method were proposed. Secondly, a TCV controller was proposed to control both the magnitude and slope of the TCV according to nonlinearly varying voltage distortion. Finally, the linear output voltage limiting regions of the three-phase VSI when the proposed DTCS is applied were defined and, also the maximum linear phase voltage magnitudes were analyzed to allow normal compensation, even at high modulation index (MI). Simulation and experiments were performed to verify the performance of the proposed DTCS, and the settable specifications of the simulation were set equal to the experimental environment. Especially, the dead-time was set as an excessive amount rather than the general condition in both the experiment and simulation to verify the proposed DTCS performance in a severe environment. Experimental results show excellent performance in wide current regions.

**Author Contributions:** J.L. designed the proposed strategy and implemented the system and performed the experiments. H.B. assisted in the research and investigation process. Y.C. assisted with the idea development and paper writing.

**Funding:** This work was supported by "Human Resources Program in Energy Technology" of the Korea Institute of Energy Technology Evaluation and Planning (KETEP), granted financial resource from the Ministry of Trade, Industry & Energy, Republic of Korea. (No. 20174030201660), and also this work was supported by the National Research Foundation of Korea (NRF) grant funded by the Korea government (MSIT) (No. 2017R1C1B2009425).

**Acknowledgments:** This work was supported by "Human Resources Program in Energy Technology" of the Korea Institute of Energy Technology Evaluation and Planning (KETEP), granted financial resource from the Ministry of Trade, Industry & Energy, Republic of Korea. (No. 20174030201660), and also this work was supported by the National Research Foundation of Korea (NRF) grant funded by the Korea government (MSIT) (No. 2017R1C1B2009425).

**Conflicts of Interest:** The authors declare no conflict of interest.

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
