# Peer review of "Novel Dead-Time Compensation Strategy for Wide Current Range in a Three-Phase Inverter"

_electronics, doi:10.3390/electronics8010092_

Round 1
Reviewer 1 Report
1) The English writing needs a thorough revision. In many points the paper is very difficult to follow.
2) The most recent references used by the authors about dead-time compensation are from 2014. More recent references (from the last 2 years) should be added and commented. The results of the authors´ proposal should be compared with those of the most recent research.
3) Why is the voltage compensation in the low current region so important? What is exactly the ‘low current region’? That should be explained in the introduction section.
4) In Figure 1 the middle point of the DC-link voltage is labelled ‘a’, in the same way as the middle point of switching leg ‘a’. This should be corrected. What point is ‘n’? Besides, in fig. 1(b) the current ‘ia’ has the same direction, no matter if it´s positive or negative. That should be corrected and made compatible with the direction of the current ‘ia’ in Fig. 1(a). Figure 1 is completely misleading.
5) The conditions of Fig. 2 should be fully clarified. The scale of the x-axis is missing, so that the fundamental frequency cannot be deduced. I understand that Fig. 2(b) depicts current ia, but it´s difficult to realize why that current is so much distorted at its peak values when the dead time value is Td=5 microseconds. A distortion of the current at its zero crossings is expected, but not its almost trapezoidal waveform. Besides, the chosen dead time value is too high when compared to the switching period (50 microseconds). In my opinion, the conditions of Fig. 2 are not realistic.
6) What is the exact definition of the error voltages Delta_vas_err, Delta_vbs_err and Delta_vcs_err in equation (3)?
7) How are the alpha/beta and the d/q axis aligned related to the phase voltages (figure 3)?
8) Please, correct the nomenclature of the switching pole voltage nodes (‘a’ is depicted twice). Is the phase a load current ‘ia’ flowing to the middle point of the DC-link?
9) Equation (12) considers that the parasitic capacitance value of the upper and lower switches is the same and also constant during the switching process, but it´s well-known that that value is highly dependent on the voltage at the switches, which changes during the switching process. Is equation (12) useful or reliable to devise any dead-time compensation strategy? In my opinion the graph representing Toff (figure 6) doesn´t make much sense. Which simulation software has been used for obtaining the value of Toff_sim? Does it take into account the non-linear behaviour of C12? Which value of C12 has been considered? The values of Toff for low current are nonsense, because they are higher than the switching period.
10) Figure 7 doesn´t agree with the real waveforms which can be measured in an inverter, because a constant value of C12 has been assumed. Besides, the scale of the X-axis is missing.
11) Which real switch has been used for getting the waveforms of Fig. 8. What value of the turn-on and turn-off gate resistors have been used? Are the switches IGBTs, Mosfets, … ?
12) The IGBT chosen for the simulation results (600 V, 50 A) is not a good choice for working with a 100 V DC-link. Unless a special workpackage is used, PSIM software is not good for simulating switching effects and the effects of parasitic capacitances. In my opinion the conclusions drawn from the simulation results are not realistic.
13) Both in the experimental prototype and in the simulations the value of the dead time is too high compared to the switching period. Why has a different value of the switching frequency been chosen in the experimental prototype (15 kHz) and in the simulations (20 kHz)?
14) In my opinion the conclusions drawn from the experimental results are not realistic, because the value of the dead time isn´t realistic either.
Author Response
Dear Reviewers,
Thank you very much for your time and effort. Your comments are highly appreciated.
The following modifications were done based on your comments:
For Reviewer 1
Comment 1 | 1) The English writing needs a thorough revision. In many points the paper is very difficult to follow. |
Answer | Thank you for your valuable comment. The paper has been revised the whole English sentence and word. In addition, the authors plan to use professional English editing service before making the final publication. |
Comment 2 | 2) The most recent references used by the authors about dead-time compensation are from 2014. More recent references (from the last 2 years) should be added and commented. The results of the authors´ proposal should be compared with those of the most recent research. |
Answer | Thank you for your opinions. The latest 2 years papers related to dead-time have been added and compared. Also, the discussion about the added papers was carried out by the introduction section.
[18] TANG, Zhuangyao; AKIN, Bilal. Suppression of dead-time distortion through revised repetitive controller in PMSM Drives. IEEE Transactions on Energy Conversion, 2017, 32.3: 918-930. [19] LIU, Gang, et al. Current-Detection-Independent Dead-Time Compensation Method Based on Terminal Voltage A/D Conversion for PWM VSI. IEEE Transactions on Industrial Electronics, 2017, 64.10: 7689-7699. [20] DAFANG, Wang, et al. A feedback-type phase voltage compensation strategy based on phase current reconstruction for ACIM drives. IEEE Trans. Power Electron., 2014, 29.9: 5031-5043. |
Comment 3 | 3) Why is the voltage compensation in the low current region so important? What is exactly the ‘low current region’? That should be explained in the introduction section. |
Answer | I agree with your question. The reason for mentioning the low current region in this paper is that the parasitic components of the switch appear in the low current region and affect the inverter output voltage. Therefore, in this paper, a strategy that actively compensates for these changes was proposed. As a result, an upper content of the low current has been added to the introduction section to reflect your opinion. The modified paper contents have been added as the following.
“In this paper, a novel DTCS for accurate dead-time compensation in all output regions of the inverter is proposed. In particular, the dead-time compensation algorithm using passive calculation for the compensation voltage is difficult to accurately compensate all areas of the inverter output because the switch turn-off delay effects increased occurs due to the parasitic components of the switch in the low current region. Therefore, in this paper, a new controller that can actively compensate for voltage distortion due to the switch parasitic components and a new method that can more easily implement the three-phase TCV are presented.” |
Comment 4 | 4) In Figure 1 the middle point of the DC-link voltage is labelled ‘a’, in the same way as the middle point of switching leg ‘a’. This should be corrected. What point is ‘n’? Besides, in fig. 1(b) the current ‘ia’ has the same direction, no matter if it´s positive or negative. That should be corrected and made compatible with the direction of the current ‘ia’ in Fig. 1(a). Figure 1 is completely misleading. |
Answer | Figure 1 of this paper refers on the following paper. I wondered why the reviewer was confused about Figure 1, and thought it was because the title of the figure was not properly represented. Figure 1 (a) shows a basic three-phase inverter, and Figure 1 (b) shows one leg of a three-phase inverter in (a). Therefore, the title of Figure 1 has been revised. If the problem with the figure is different from what I understand, please give me further feedbacks.
Reference paper: Tang, Zhuangyao, and Bilal Akin. "Suppression of dead-time distortion through revised repetitive controller in PMSM Drives." IEEE Transactions on Energy Conversion 32.3 (2017): 918-930. |
Comment 5 | 5) The conditions of Fig. 2 should be fully clarified. The scale of the x-axis is missing, so that the fundamental frequency cannot be deduced. I understand that Fig. 2(b) depicts current ia, but it´s difficult to realize why that current is so much distorted at its peak values when the dead time value is Td=5 microseconds. A distortion of the current at its zero crossings is expected, but not its almost trapezoidal waveform. Besides, the chosen dead time value is too high when compared to the switching period (50 microseconds). In my opinion, the conditions of Fig. 2 are not realistic. |
Answer | This is very important comment for the readers. First, the figure 2 was added the x-axis scale. Second, the reason for setting the dead-time to 5 micro-second is to maximize the current distortion due to the dead-time. The dead-time size has differences of the distortion magnitudes but not the functional mis-operation. Third, the reason why the phase current distortion due to the dead-time occurs at the peak area is because the three-phase load is connected to Y as shown in Fig. 1, and the current distortion generated at other phases affects the phase current. Therefore, we modified the title of Figure 2 to help understand the load connection. |
Comment 6 | 6) What is the exact definition of the error voltages Delta_vas_err, Delta_vbs_err and Delta_vcs_err in equation (3)? |
Answer | Thank you for the comment. Equation (3) is the value obtained by converting the polar voltage error of Equation (1) into the phase voltage error. I added a reference to it because it is a formula already defined in the reference. The following are the modifications:
“Equation (3) is expressed as the average phase voltage error by APVE [3], and Equation (4) represents the average phase voltage error due to the dead-time on the synchronous reference frame d-q axis by the Fourier series expansion [5] [11].” |
Comment 7 | 7) How are the alpha/beta and the d/q axis aligned related to the phase voltages (figure 3)? |
Answer | To illustrate the relationship between the phase voltage and the alpha / beta and d / q axes in Figure 3, a-phase was additionally shown. The phases of the alpha axis and a phase are the same phase. The modified figure is follows. |
Comment 8 | 8) Please, correct the nomenclature of the switching pole voltage nodes (‘a’ is depicted twice). Is the phase a load current ‘ia’ flowing to the middle point of the DC-link? |
Answer | Thank you for your considerations. The 'a' node was changed to the 'p' node in Figure 5 and modified all the variables and equations accordingly. Also, confusions have been prevented by changing the current 'ia' flowing to the n-node mentioned above to 'ip'. |
Comment 9 | 9) Equation (12) considers that the parasitic capacitance value of the upper and lower switches is the same and also constant during the switching process, but it´s well-known that that value is highly dependent on the voltage at the switches, which changes during the switching process. Is equation (12) useful or reliable to devise any dead-time compensation strategy? In my opinion the graph representing Toff (figure 6) doesn´t make much sense. Which simulation software has been used for obtaining the value of Toff_sim? Does it take into account the non-linear behaviour of C12? Which value of C12 has been considered? The values of Toff for low current are nonsense, because they are higher than the switching period. |
Answer | Section 2.2 of this paper has a purpose that correct dead-time compensation is not possible using only fixed variables. Therefore, equation (12) is a formula shows that the delay time Toff changes according to the current magnitude. In this paper, also, it is very unlikely that a voltage change that can affects Toff is very rare because it is aimed at applications that maintain a constant voltage level (e.g., grid-connected inverters, motor drive systems). Therefore, assuming that the capacitance is a fixed variable, it is reasonable to reflect the reality. The simulation results in this paper have used Psim. Capacitor C12 is a fixed variable and it utilizes the output capacitance indicated in the data sheet of the target switch; SKM50GB063D. Also, since the bottom switch keeps turned off, only the top switch is turned on / off so that the data shown in Fig. 6 can be obtained. These are added to the text as follows.
“Figures 6 and 7 show the simulation waveforms and the results to verify the previously defined equations in section 2.2. The graph of Figure 6 is comparing the simulation results of Figure 7 with equation (12) where is the turn off delay times measured using the simulation results in Figure 7, and is the turn off delay times calculated using the equation (12) respectively. In here, the simulation circuit configuration of Figure 7 is arranged such as Figure 5 (a), and the lower switch is keeping turning off condition while upper switch is turning on and off. Figure 7 (a) shows the gate-source voltage of the upper switch, and the lower switch waveform is omitted because it only applies the off signal. Figure 7 (c) displays the pole voltage and voltage distortion due to the output capacitor can be confirmed.”
|
Comment 10 | 10) Figure 7 doesn´t agree with the real waveforms which can be measured in an inverter, because a constant value of C12 has been assumed. Besides, the scale of the X-axis is missing. |
Answer | Figure 7 shows the waveforms of the simulation result and Figure 8 shows the experimental waveform in order to provide the basis for the waveforms in Figure 7. Thus, I believe that the correct of Figure 7 is can be explained through Figure 8. |
Comment 11 | 11) Which real switch has been used for getting the waveforms of Fig. 8. What value of the turn-on and turn-off gate resistors have been used? Are the switches IGBTs, Mosfets, … ? |
Answer | Thank you for the comment. The waveform in Figure 8 uses the SKM50GB063D IGBT, and details have been added to the figure title. |
Comment 12 | 12) The IGBT chosen for the simulation results (600 V, 50 A) is not a good choice for working with a 100 V DC-link. Unless a special workpackage is used, PSIM software is not good for simulating switching effects and the effects of parasitic capacitances. In my opinion the conclusions drawn from the simulation results are not realistic. |
Answer | This is critical comment, thank you for your delicacy. In fact, the DC-link voltage level was set at 100V because it was a personal setting to compare the magnitude of the compensation voltage amplitudes intuitively with the DC-link level. Therefore, the simulation results were corrected by setting the DC-link voltage level and switching frequency to the equal conditions as the experimental environment. Also, as I answer above, the capacitance of variation is no problem because the application (motor drive) assume that the DC-link voltage is fixed. |
Comment 13 | 13) Both in the experimental prototype and in the simulations the value of the dead time is too high compared to the switching period. Why has a different value of the switching frequency been chosen in the experimental prototype (15 kHz) and in the simulations (20 kHz)? |
Answer | There has been an inevitable change due to the experimental environment constraints. Therefore, the simulation results were modified to the equal conditions as the experimental environment. |
Comment 14 | 14) In my opinion the conclusions drawn from the experimental results are not realistic, because the value of the dead time isn´t realistic either. |
Answer | Thank you for your considerations. In fact, I also have experimental results on 1 micro-second and I have found that the algorithm also works well in this environment. But at 1micro-second, the dead time is hardly visible due to the delay of the switch turn off, so I the experiment with 5 micro-second and the waveforms are thought to provide the reader with more understanding and information. The experimental results at 5 micro-second is the worse experimental condition because the size of the dead time only determines the magnitude of the error voltage. And the proposed algorithm is working well in that environment. Therefore, I added reason in conclusion section why I selected dead-time to 5 micro-second instead of displaying 1 micro-second results. If you do not agree with my opinion, I would appreciate any further comments. the result waveforms of 1 micro-second are below. Conventional dead-time compensation algorithm Without any dead-time compensation algorithm With proposed dead-time compensation strategy |
Best Regards,
Authors

Reviewer 2 Report
The paper is well developed and the authors provide much information regarding the proposed approach.
However, the significance must be improved, choosing 5 us of deadtime for the experiments is not realistic, please also add a comparison between the proposed method and the state of the art with reduced deadtime (1 us).
Author Response
Dear Reviewers,
Thank you very much for your time and effort. Your comments are highly appreciated.
The following modifications were done based on your comments:
For Reviewer 1
Comment 1 | The paper is well developed and the authors provide much information regarding the proposed approach. However, the significance must be improved, choosing 5 us of deadtime for the experiments is not realistic, please also add a comparison between the proposed method and the state of the art with reduced deadtime (1 us). |
Answer | Thank you for your considerations. In fact, I also have experimental results on 1 micro-second and I have found that the algorithm also works well in this environment. But at 1micro-second, the dead time is hardly visible due to the delay of the switch turn off, so I the experiment with 5 micro-second and the waveforms are thought to provide the reader with more understanding and information. The experimental results at 5 micro-second is the worse experimental condition because the size of the dead time only determines the magnitude of the error voltage. And the proposed algorithm is working well in that environment. Therefore, I added reason in conclusion section why I selected dead-time to 5 micro-second instead of displaying 1 micro-second results. If you do not agree with my opinion, I would appreciate any further comments. the result waveforms of 1 micro-second are below. Conventional dead-time compensation algorithm Without any dead-time compensation algorithm With proposed dead-time compensation strategy |
Best Regards,
Authors

Reviewer 3 Report
A lot of work is done but the presentation is not good enough. The English must be fixed. It is too hard to understand your explanations. There are some small errors too, like missing letters, wrong-written words etc. The abbreviations need to be defined.
Author Response
Dear Reviewers,
Thank you very much for your time and effort. Your comments are highly appreciated.
The following modifications were done based on your comments:
For Reviewer 1
Comment 1 | A lot of work is done but the presentation is not good enough. The English must be fixed. It is too hard to understand your explanations. There are some small errors too, like missing letters, wrong-written words etc. The abbreviations need to be defined. |
Answer | Thank you for your valuable comment. According to your guidance, the use of English language has been carefully revised in this submission. The expressions you have mentioned have been corrected, and articles have been double-checked. The entire paragraphs have been also reviewed and corrected. In addition, the authors plan to use professional English editing service before making the final publication. |
Best Regards,
Authors

Round 2
Reviewer 1 Report
1) After reading the authors´answer to my comments of the first version of the manuscript, I understand that a wrong concept persists. The values of parasitic capacitances C1 and C2 (Coss of the IGBT or mosfet transistors) are not constant during the switching process, even if the DC link voltage is constant, because those capacitances depend on the voltage at the switch, which is changing during turn-on and turn-off. In the case of IGBT transistors, the switching process is even more complicated than in the case of mosfets during turn-off, because of the 'current tail' effect. Therefore, some conclusions of the paper, like equations (10) to (12), are wrong.
2) A dead time of 5 microseconds is still not realistic (too high) for a switching frequency of 10 kHz, the new value chosen by the authors in version V2 of the manuscript. Therefore, I don´t think that the improvement in terms of THD reached by the proposed method is as good as it seems in the paper: the THD without dead-time compensation is worse than it would be with a proper value of dead-time.
Author Response
Dear Reviewers,
Thank you very much for your time and effort. Your comments are highly appreciated.
The following modifications were done based on your comments:
For Reviewer 1
Comment 1 | 1) After reading the authors´answer to my comments of the first version of the manuscript, I understand that a wrong concept persists. The values of parasitic capacitances C1 and C2 (Coss of the IGBT or mosfet transistors) are not constant during the switching process, even if the DC link voltage is constant, because those capacitances depend on the voltage at the switch, which is changing during turn-on and turn-off. In the case of IGBT transistors, the switching process is even more complicated than in the case of mosfets during turn-off, because of the 'current tail' effect. Therefore, some conclusions of the paper, like equations (10) to (12), are wrong. |
Answer | Thank you for your considerations. In fact, as the reviewer said, there may be a change in capacitance. However, many papers that have published research on the dead-time turn-off delay have done not mention the change in capacitance due to the voltage. I raise that the effects of capacitance changes are negligible or that there is little or no change in capacitance. The following papers deal with the effects of switch output capacitors on inverter output and have conclusions similar to this paper. please check the following papers.
A. Guha and G. Narayanan, "An improved dead-time compensation scheme for voltage source inverters considering the device switching transition times," in Power Electronics (IICPE), 2014 IEEE 6th India International Conference on, 2014, pp. 1-6: IEEE.
N. Urasaki, T. Senjyu, T. Kinjo, T. Funabashi, and H. Sekine, "Dead-time compensation strategy for permanent magnet synchronous motor drive taking zero-current clamp and parasitic capacitance effects into account," IEE Proceedings - Electric Power Applications, vol. 152, no. 4, pp. 845-853, 2005.
Y. Park and S.-K. Sul, "A novel method utilizing trapezoidal voltage to compensate for inverter nonlinearity," IEEE Transactions on power Electronics, vol. 27, no. 12, pp. 4837-4846, 2012.
T. Mannen and H. Fujita, "Dead-Time Compensation Method Based on Current Ripple Estimation," IEEE Transactions on Power Electronics, vol. 30, no. 7, pp. 4016-4024, 2015. G. Liu, D. Wang, Y. Jin, M. Wang, and P. Zhang, "Current-Detection-Independent Dead-Time Compensation Method Based on Terminal Voltage A/D Conversion for PWM VSI," IEEE Transactions on Industrial Electronics, vol. 64, no. 10, pp. 7689-7699, 2017.
Z. Zhang and L. Xu, "Dead-time compensation of inverters considering snubber and parasitic capacitance," IEEE Transactions on Power Electronics, vol. 29, no. 6, pp. 3179-3187, 2014. |
Comment 2 | 2) A dead time of 5 microseconds is still not realistic (too high) for a switching frequency of 10 kHz, the new value chosen by the authors in version V2 of the manuscript. Therefore, I don´t think that the improvement in terms of THD reached by the proposed method is as good as it seems in the paper: the THD without dead-time compensation is worse than it would be with a proper value of dead-time. |
Answer | I totally agree with the opinion that there is no reality. However, the reason why the dead-time is set to 5usec, as mentioned previously, to help the reader more information and understanding. Also, as suggested by the reviewer, the proposed dead-time compensation strategy performed on 1usec and it has been completed (including the waveform on the prior cover letter), so the proposed strategy does not operate in only specific area. The following papers for dead-time compensation had been set a dead-time of at least 3 usec and even up to 12.5usec. I believe this is a setting value to show the performance rather than the reality and this paper also sets the dead-time size for this purpose. Therefore, the experimental environment and experimental results presented in this paper have no problem what the readers have been confirming the performance of the proposed dead-time compensation algorithm.
D. Lee and J. Ahn, "A Simple and Direct Dead-Time Effect Compensation Scheme in PWM-VSI," IEEE Transactions on Industry Applications, vol. 50, no. 5, pp. 3017-3025, 2014.
S.-Y. Kim, W. Lee, M.-S. Rho, and S.-Y. Park, "Effective dead-time compensation using a simple vectorial disturbance estimator in PMSM drives," IEEE Transactions on Industrial Electronics, vol. 57, no. 5, pp. 1609-1614, 2010.
C. Attaianese and G. Tomasso, "Predictive compensation of dead-time effects in VSI feeding induction motors," IEEE Transactions on Industry Applications, vol. 37, no. 3, pp. 856-863, 2001.
M. El-daleel and A. Mahgoub, "Accurate and simple improved lookup table compensation for inverter dead time and nonlinearity compensation," in Power Systems Conference (MEPCON), 2017 Nineteenth International Middle East, 2017, pp. 1358-1361: IEEE.
C. Choi, K. Cho, and J. Seok, "Inverter Nonlinearity Compensation in the Presence of Current Measurement Errors and Switching Device Parameter Uncertainties," IEEE Transactions on Power Electronics, vol. 22, no. 2, pp. 576-583, 2007.
Z. Zhang and L. Xu, "Dead-time compensation of inverters considering snubber and parasitic capacitance," IEEE Transactions on Power Electronics, vol. 29, no. 6, pp. 3179-3187, 2014. |
Best Regards,
Authors

Reviewer 3 Report
This is a result of a huge work, but it looks more like a thesis or some technical report, than a paper. Too heavy, with too many details, too many words (and this is not for the better). I would suggest the following:
The objective of each chapter should be presented shortly and clearly, all variables in equations should be defined.
There are no need to explain what is done in equation, a reader can see it. If needed, explanations can be put in the appendix.
Axes in plots should be defined clearly. It is difficult to separate one compensation voltage from another compensation voltage and another compensation voltage, all on the same plot.
Please, check Fig. 13. What are the units on x-axes there?
English is readable now but needs correction.
A lot of work is done, good results are achieved. All this should be presented correctly.
Author Response
I uploaded word for cover letter, please check the attached file.
